# Minimum width for universal approximation using ReLU networks on compact domain

**Namjun Kim**[1]    **Chanho Min**[2]    **Sejun Park**[1*]
[1]Korea University    [2]Ajou University

## Abstract

It has been shown that deep neural networks of a large enough width are universal approximators but they are not if the width is too small. There were several attempts to characterize the minimum width $w_{\min}$ enabling the universal approximation property; however, only a few of them found the exact values. In this work, we show that the minimum width for $L^p$ approximation of $L^p$ functions from $[0, 1]^{d_x}$ to $\mathbb{R}^{d_y}$ is exactly $\max\{d_x, d_y, 2\}$ if an activation function is ReLU-Like (e.g., ReLU, GELU, Softplus). Compared to the known result for ReLU networks, $w_{\min} = \max\{d_x + 1, d_y\}$ when the domain is $\mathbb{R}^{d_x}$, our result first shows that approximation on a compact domain requires smaller width than on $\mathbb{R}^{d_x}$. We next prove a lower bound on $w_{\min}$ for uniform approximation using general activation functions including ReLU: $w_{\min} \geq d_y + 1$ if $d_x < d_y \leq 2d_x$. Together with our first result, this shows a dichotomy between $L^p$ and uniform approximations for general activation functions and input/output dimensions.

## 1 Introduction

Understanding what neural networks can or cannot do is a fundamental problem in the expressive power of neural networks. Initial approaches for this problem mostly focus on depth-bounded networks. For example, a line of research studies the size of the two-layer neural network to memorize (i.e., perfectly fit) an arbitrary training dataset and shows that the number of parameters proportional to the dataset size is necessary and sufficient for various activation functions (Baum, 1988; Huang and Babri, 1998). Another important line of work investigates a class of functions that two-layer networks can approximate. Classical results in this field represented by the universal approximation theorem show that two-layer networks using a non-polynomial activation function are dense in the space of continuous functions on compact domains (Hornik et al., 1989; Cybenko, 1989; Leshno et al., 1993; Pinkus, 1999).

With the success of deep learning, the expressive power of deep neural networks has been studied. As in the classical depth-bounded network results, several works have shown that width-bounded networks can memorize arbitrary training dataset (Yun et al., 2019; Vershynin, 2020) and can approximate any continuous function (Lu et al., 2017; Hanin and Sellke, 2017). Intriguingly, it has also been shown that deeper networks can be more expressive compared to shallow ones. For example, Telgarsky (2016); Eldan and Shamir (2016); Daniely (2017) show that there is a class of functions that can be approximated by deep width-bounded networks with a small number of parameters but cannot be approximated by shallow networks without extremely large widths. Furthermore, width-bounded networks require a smaller number of parameters for universal approximation (Yarotsky, 2018) and memorization (Park et al., 2021a; Vardi et al., 2022) compared to depth-bounded ones.

Recently, researchers started to identify the *minimum width* that enables universal approximation of width-bounded networks as a dual problem of the classical results: the minimum depth of neural networks for universal approximation is *exactly two* if their activation function is non-polynomial. Unlike the minimum depth independent of the input dimension $d_x$ and the output dimension $d_y$ of target functions, the minimum width is known to lie between $d_x$ and $d_x + d_y + \alpha$ for various activation functions where $\alpha$ is some non-negative number depending on the activation function (Lu et al., 2017; Hanin and Sellke, 2017; Johnson, 2019; Kidger and Lyons, 2020; Park et al., 2021b; Cai, 2023). However, most existing results only provide bounds on the minimum width, and the exact minimum width is known for a few activation functions and problem setups so far.

---

*corresponding author
    emails: namjun-kim@korea.ac.kr, chanhomin@ajou.ac.kr, sejun.park000@gmail.com

Table 1: A summary of known bounds on the minimum width for universal approximation. In this table, $p \in [1, \infty)$ and all results with the domain $[0,1]^{d_x}$ extends to an arbitrary compact set in $\mathbb{R}^{d_x}$.

| Reference | Function class | Activation $\sigma$ | Upper/lower bounds |
|---|---|---|---|
| Lu et al. (2017) | $L^1(\mathbb{R}^{d_x}, \mathbb{R})$ | RELU | $d_x + 1 \leq w_{\min} \leq d_x + 4$ |
| | $L^1([0,1]^{d_x}, \mathbb{R})$ | RELU | $w_{\min} \geq d_x$ |
| Hanin and Sellke (2017) | $C([0,1]^{d_x}, \mathbb{R}^{d_y})$ | RELU | $d_x + 1 \leq w_{\min} \leq d_x + d_y$ |
| Johnson (2019) | $C([0,1]^{d_x}, \mathbb{R}^{d_y})$ | uniformly conti.† | $w_{\min} \geq d_x + 1$ |
| Kidger and Lyons (2020) | $C([0,1]^{d_x}, \mathbb{R}^{d_y})$ | conti. nonpoly.‡ | $w_{\min} \leq d_x + d_y + 1$ |
| | $C([0,1]^{d_x}, \mathbb{R}^{d_y})$ | nonaffine poly. | $w_{\min} \leq d_x + d_y + 2$ |
| | $L^p(\mathbb{R}^{d_x}, \mathbb{R}^{d_y})$ | RELU | $w_{\min} \leq d_x + d_y + 1$ |
| Park et al. (2021b) | $L^p(\mathbb{R}^{d_x}, \mathbb{R}^{d_y})$ | RELU | $w_{\min} = \max\{d_x + 1, d_y\}$ |
| | $C([0,1], \mathbb{R}^2)$ | RELU | $w_{\min} > \max\{d_x + 1, d_y\}$ |
| | $L^p([0,1]^{d_x}, \mathbb{R}^{d_y})$ | conti. nonpoly.‡ | $w_{\min} \leq \max\{d_x + 2, d_y + 1\}$ |
| Cai (2023) | $L^p([0,1]^{d_x}, \mathbb{R}^{d_y})$ | Leaky-RELU | $w_{\min} = \max\{d_x, d_y, 2\}$ |
| | $L^p([0,1]^{d_x}, \mathbb{R}^{d_y})$ | arbitrary | $w_{\min} \geq \max\{d_x, d_y\}$ |
| | $C([0,1]^{d_x}, \mathbb{R}^{d_y})$ | arbitrary | $w_{\min} \geq \max\{d_x, d_y\}$ |
| **Ours** (Theorem 1) | $L^p([0,1]^{d_x}, \mathbb{R}^{d_y})$ | RELU | $w_{\min} = \max\{d_x, d_y, 2\}$ |
| **Ours** (Theorem 2) | $L^p([0,1]^{d_x}, \mathbb{R}^{d_y})$ | RELU-LIKE§ | $w_{\min} = \max\{d_x, d_y, 2\}$ |
| **Ours** (Theorem 3) | $C([0,1]^{d_x}, \mathbb{R}^{d_y})$ | conti.† | $w_{\min} \geq d_y + \mathbf{1}_{d_x < d_y \leq 2d_x}$ |

† requires that $\sigma$ is uniformly approximated by a sequence of continuous one-to-one functions.
‡ requires that $\sigma$ is continuously differentiable at least one point $z$, with $\sigma'(z) \neq 0$.
§ includes SOFTPLUS, Leaky-RELU, ELU, CELU, SELU, GELU, SILU, and MISH where GELU, SILU, and MISH require $d_x + d_y \geq 3$.

## 1.1 RELATED WORKS

Before summarizing prior works, we first define function spaces often considered in universal approximation literature. We use $C(\mathcal{X}, \mathcal{Y})$ to denote the space of all continuous functions from $\mathcal{X} \subset \mathbb{R}^{d_x}$ to $\mathcal{Y} \subset \mathbb{R}^{d_y}$, endowed with the uniform norm: $\|f\|_\infty \triangleq \sup_{x \in \mathcal{X}} \|f(x)\|_\infty$. We also define $L^p(\mathcal{X}, \mathcal{Y})$ for denoting the $L^p$ space, i.e., the class of all functions with finite $L^p$-norm, endowed with the $L^p$-norm: $\|f\|_p \triangleq (\int_{\mathcal{X}} \|f\|_p^p d\mu_{d_x})^{1/p}$ where $\mu_{d_x}$ denotes the $d_x$-dimensional Lebesgue measure. We denote the minimum width for universal approximation by $w_{\min}$. See Section 2 for more detailed problem setup. Under these notations, Table 1 summarizes the known upper and lower bounds on the minimum width for universal approximation under various problem setups.

**Initial approaches.** Lu et al. (2017) provide the first upper bound $w_{\min} \leq d_x + 4$ for universal approximation of $L^1(\mathbb{R}^{d_x}, \mathbb{R})$ using (fully-connected) RELU networks. They explicitly construct a network of width $d_x + 4$ which approximates a target $L^1$ function by using $d_x$ neurons to store the $d_x$-dimensional input, one neuron to transfer intermediate constructions of the one-dimensional output, and the remaining three neurons to compute iterative updates of the output. For multi-dimensional output cases, similar constructions storing the $d_x$-dimensional input and $d_y$-dimensional (intermediate) outputs are used to prove upper bounds on $w_{\min}$ under various problem setups. For example, Hanin and Sellke (2017) show that RELU networks of width $d_x + d_y$ are dense in $C([0,1]^{d_x}, \mathbb{R}^{d_y})$. Kidger and Lyons (2020) also prove upper bounds on the minimum width for general activation functions using similar constructions. They prove $w_{\min} \leq d_x + d_y + 1$ for $C([0,1]^{d_x}, \mathbb{R}^{d_y})$ if an activation function $\sigma$ is non-polynomial and $\sigma'(z) \neq 0$ for some $z$, $w_{\min} \leq d_x + d_y + 2$ for $C([0,1]^{d_x}, \mathbb{R}^{d_y})$ if $\sigma$ is non-affine polynomial, and $w_{\min} \leq d_x + d_y + 1$ for $L^p(\mathbb{R}^{d_x}, \mathbb{R}^{d_y})$ if $\sigma = $ RELU.

Lower bounds on the minimum width have also been studied. For RELU networks, Lu et al. (2017) show that the minimum width is at least $d_x + 1$ and $d_x$ to universally approximate $L^p(\mathbb{R}^{d_x}, \mathbb{R}^{d_y})$ and $L^p([0,1]^{d_x}, \mathbb{R}^{d_y})$, respectively. Johnson (2019) considers general activation functions and shows that the minimum width is at least $d_x + 1$ to universally approximate $C([0,1]^{d_x}, \mathbb{R}^{d_y})$ if an activation function is uniformly continuous and can be uniformly approximated by a sequence of continuous and one-to-one functions. However, since these upper and lower bounds have a large gap of at least $d_y - 1$, they could not achieve the tight minimum width.

**Recent progress.** Recently, Park et al. (2021b) characterize the exact minimum width of RELU networks for universal approximation of $L^p(\mathbb{R}^{d_x}, \mathbb{R}^{d_y})$: $w_{\min} = \max\{d_x + 1, d_y\}$. To bypass width $d_x + d_y$ in the previous constructions and to prove the tight upper bound $w_{\min} \le \max\{d_x + 1, d_y\}$, they proposed the coding scheme which first encodes a $d_x$-dimensional input $x$ to a scalar-valued codeword $c$ and decodes that codeword to a $d_y$-dimensional vector approximating $f(x)$. They approximate each of these functions using RELU networks of width $d_x + 1$, and $\max\{d_y, 2\}$ as in previous constructions, which results in the tight upper bound. Using a similar construction, they also prove that networks of width $\max\{d_x + 2, d_y + 1\}$ are dense in $L^p([0, 1]^{d_x}, \mathbb{R}^{d_y})$ if an activation function $\sigma$ is continuous, non-polynomial, and $\sigma'(z) \ne 0$ for some $z \in \mathbb{R}$. For universal approximation of $L^p([0, 1]^{d_x}, \mathbb{R}^{d_y})$ using Leaky-RELU networks, Cai (2023) characterizes $w_{\min} = \max\{d_x, d_y, 2\}$ using the results that continuous $L^p$ functions can be approximated by neural ordinary differential equations (ODEs) (Li et al., 2022) and narrow Leaky-RELU networks can approximate neural ODEs (Duan et al., 2022). However, except for these two cases, the exact minimum width for universal approximation is still unknown.

One interesting observation made by Park et al. (2021b) is that RELU networks of width 2 are dense in $L^p(\mathbb{R}, \mathbb{R}^2)$ but not dense in $C([0, 1], \mathbb{R}^2)$. This shows a gap between minimum widths for $L^p$ and uniform approximations. Cai (2023) also suggests a similar dichotomy for leaky-RELU networks when $d_x = 1$ and $d_y = 2$. Nevertheless, whether such a dichotomy exists for general activation functions and input/output dimensions is unknown.

## 1.2 SUMMARY OF RESULTS

In this work, we primarily focus on characterizing the minimum width of fully-connected RELU networks for universal approximation on a compact domain. However, our results are not restricted to RELU; they extend to general activation functions as summarized below.

- Theorem 1 states that width $\max\{d_x, d_y, 2\}$ is necessary and sufficient for RELU networks to be dense in $L^p([0, 1]^{d_x}, \mathbb{R}^{d_y})$. Compared to the existing result that the minimum width is $\max\{d_x + 1, d_y\}$ when the domain is $\mathbb{R}^{d_x}$ (Park et al., 2021b), our result shows a gap between minimum widths for $L^p$ approximation on the compact and unbounded domains. To our knowledge, this is the first result showing such a dichotomy.

- Given the exact minimum width in Theorem 1, our next result shows that the same $w_{\min}$ holds for the networks using any of RELU-LIKE activation functions. Specifically, Theorem 2 states that width $\max\{d_x, d_y, 2\}$ is necessary and sufficient for $\sigma$ networks to be dense in $L^p([0, 1]^{d_x}, \mathbb{R}^{d_y})$ if $\sigma$ is in {SOFTPLUS, Leaky-RELU, ELU, CELU, SELU}, or $\sigma$ is in {GELU, SILU, MISH} and $d_x + d_y \ge 3$, which generalizes the previous result for Leaky-RELU networks (Cai, 2023).

- Our last result improves the previous lower bound on the minimum width for uniform approximation: $w_{\min} \ge d_y + 1$ for RELU networks if $d_x = 1, d_y = 2$. Theorem 3 states that $\sigma$ networks of width $d_y$ is *not dense* in $C([0, 1]^{d_x}, \mathbb{R}^{d_y})$ if $d_x < d_y \le 2d_x$ and $\sigma$ can be uniformly approximated by a sequence of continuous injections, e.g., monotone functions such as RELU.

- For uniform approximation using Leaky-RELU networks, the lower bound in Theorem 3 is tight if $d_y = 2d_x$: there is a matching upper bound $\max\{2d_x + 1, d_y\}$ (Hwang, 2023). Furthermore, together with Theorems 1 and 2, Theorem 3 extends the prior observations showing the dichotomy between $L^p$ and uniform approximations (Park et al., 2021b; Cai, 2023) to general activation functions and input/output dimensions.

- Our proof techniques also generalize to $L^p$ approximation of sequence-to-sequence functions via recurrent neural networks (RNNs). Theorems 24 and 25 in Appendix F show that the same $w_{\min}$ in Theorems 1 and 2 holds for RNNs. In addition, Theorem 26 in Appendix F shows that $w_{\min} \le \max\{d_x, d_y, 2\}$ for bidirectional RNNs using RELU or RELU-LIKE activation functions.

## 1.3 ORGANIZATION

We first introduce notations and our problem setup in Section 2. In Section 3, we formally present our main results and discuss them. In Section 4, we present the proof of the tight upper bound on the minimum width in Theorem 1. In Section 5, we prove Theorem 3 by providing a continuous function $f^* : \mathbb{R}^{d_x} \to \mathbb{R}^{d_y}$ with $d_x < d_y \le 2d_x$ that cannot be uniformly approximated by a width-$d_y$ network using general activation functions. Lastly, we conclude the paper in Section 6.

## 2  PROBLEM SETUP AND NOTATION

We mainly consider fully-connected neural networks that consist of affine transformations and an activation function. Given an activation function $\sigma : \mathbb{R} \to \mathbb{R}$, we define an $L$-layer neural network $f$ of input and output dimensions $d_x, d_y \in \mathbb{N}$, and hidden layer dimensions $d_1, \ldots, d_{L-1}$ as follows:

$$f \triangleq t_L \circ \phi_{L-1} \circ \cdots \circ t_2 \circ \phi_1 \circ t_1, \tag{1}$$

where $t_\ell : \mathbb{R}^{d_{\ell-1}} \to \mathbb{R}^{d_\ell}$ is an affine transformation and $\phi_\ell$ is defined as $\phi_\ell(x_1, \ldots, x_{d_\ell}) = (\sigma(x_1), \ldots, \sigma(x_{d_\ell}))$ for all $\ell \in [L]$. We denote a neural network $f$ with an activation function $\sigma$ by a "$\sigma$ network." We define the width of $f$ as the maximum over $d_1, \ldots, d_{L-1}$.

We say "$\sigma$ networks of width $w$ are dense in $C(\mathcal{X}, \mathcal{Y})$" if for any $f^* \in C(\mathcal{X}, \mathcal{Y})$ and $\varepsilon > 0$, there exists a $\sigma$ network $f$ of width $w$ such that $\|f^* - f\|_\infty \le \varepsilon$. Likewise, we say $\sigma$ networks of width $w$ are dense in $L^p(\mathcal{X}, \mathcal{Y})$ if for any $f^* \in L^p(\mathcal{X}, \mathcal{Y})$ and $\varepsilon > 0$, there exists a $\sigma$ network $f$ of width $w$ such that $\|f^* - f\|_p \le \varepsilon$. We say "$w_{\min} = w$ for $\sigma$ networks to be dense in $C(\mathcal{X}, \mathcal{Y})$ (or $L^p(\mathcal{X}, \mathcal{Y})$)" if $\sigma$ networks of width $w$ are dense in $C(\mathcal{X}, \mathcal{Y})$ (or $L^p(\mathcal{X}, \mathcal{Y})$) but $\sigma$ networks of width $w - 1$ are not. In other words, $w_{\min}$ denotes the width of neural networks necessary and sufficient for universal approximation in $C(\mathcal{X}, \mathcal{Y})$ (or $L^p(\mathcal{X}, \mathcal{Y})$).

We lastly introduce frequently used notations. For $n \in \mathbb{N}$, we use $\mu_n$ to denote the $n$-dimensional Lebesgue measure and $[n] \triangleq \{1, \ldots, n\}$. For $n \in \mathbb{N}$ and $\mathcal{S} \subset \mathbb{R}^n$, we use $\mathrm{diam}(\mathcal{S}) \triangleq \sup_{x,y \in \mathcal{S}} \|x - y\|_2$. A set $\mathcal{H}^+ \subset \mathbb{R}^n$ is a half-space if $\mathcal{H}^+ = \{x \in \mathbb{R}^n : a^\top x + b \ge 0\}$ for some $a \in \mathbb{R}^n \setminus \{(0, \ldots, 0)\}$ and $b \in \mathbb{R}$. A set $\mathcal{P} \subset \mathbb{R}^n$ is a (convex) polytope if $\mathcal{P}$ is bounded and can be represented as an intersection of finite half-spaces. For $f : \mathbb{R}^n \to \mathbb{R}^m$, $f(x)_i$ denotes the $i$-th coordinate of $f(x)$. We define RELU-LIKE activation functions (RELU, Leaky-RELU, GELU, SILU, MISH, SOFTPLUS, ELU, CELU, SELU) in Appendix A. For an activation function with parameters (e.g., Leaky-RELU and SOFTPLUS), we assume that a single parameter configuration is shared across all activation functions and it is fixed, i.e., we do not tune them when approximating a target function.

## 3  MAIN RESULTS

We are now ready to introduce our main results on the minimum width for universal approximation.

$L^p$ **approximation with RELU-LIKE activation functions.** Our first result exactly characterizes the minimum width for universal approximation of $L^p([0,1]^{d_x}, \mathbb{R}^{d_y})$ using RELU networks.

**Theorem 1.** $w_{\min} = \max\{d_x, d_y, 2\}$ *for* RELU *networks to be dense in* $L^p([0,1]^{d_x}, \mathbb{R}^{d_y})$.

Theorem 1 states that for RELU networks, width $\max\{d_x, d_y, 2\}$ is necessary and sufficient for universal approximation of $L^p([0,1]^{d_x}, \mathbb{R}^{d_y})$. Compared to the existing result that RELU networks of width $\max\{d_x, d_y, 2\}$ are not dense in $L^p(\mathbb{R}^{d_x}, \mathbb{R}^{d_y})$ if $d_x + 1 > d_y \ge 2$ (Park et al., 2021b), Theorem 1 shows a discrepancy between approximating $L^p$ functions on a compact domain (i.e., $[0,1]^{d_x}$) and on the whole Euclidean space (i.e., $\mathbb{R}^{d_x}$). Namely, a smaller width is sufficient for approximating $L^p$ functions on a compact domain if $d_x + 1 > d_y$. We note that a similar result was already known for the *minimum depth* analysis of RELU networks: two-layer RELU networks are dense in $L^p([0,1]^{d_x}, \mathbb{R}^{d_y})$ but not dense in $L^p(\mathbb{R}^{d_x}, \mathbb{R}^{d_y})$ (Lu, 2021; Wang and Qu, 2022).

Although Theorem 1 extends the result of (Cai, 2023) from Leaky-RELU networks to RELU ones, we use a completely different approach for proving the upper bound. Cai (2023) approximates a target function via a Leaky-RELU network of width $\max\{d_x, d_y, 2\}$ using two steps: approximate the target function by a neural ODE first (Li et al., 2022), then approximate the neural ODE by a Leaky-RELU network (Duan et al., 2022). Here, the latter step requires the strict monotonicity of Leaky-RELU and does not generalize to non-strictly monotone activation functions (e.g., RELU).

To bypass this issue, we carefully analyze the properties of RELU networks and propose a different construction. In particular, our construction of a RELU network of width $\max\{d_x, d_y, 2\}$ that approximates a target function is based on the coding scheme consisting of two functions: an encoder and a decoder. First, an encoder encodes each input to a scalar-valued codeword, and a decoder maps each codeword to an approximate target value. Park et al. (2021b) approximate the encoder with the domain $\mathbb{R}^{d_x}$ using a RELU network of width $d_x + 1$ and implemented the decoder using a RELU network of width $\max\{d_y, 2\}$ to obtain a universal approximator of width $\max\{d_x + 1, d_y\}$

for $L^p(\mathbb{R}^{d_x}, \mathbb{R}^{d_y})$. By exploiting the compactness of the domain and based on the functionality of RELU networks, we successfully approximate the encoder using a RELU network of width $\max\{d_x, 2\}$ and show that RELU networks of width $\max\{d_x, d_y, 2\}$ is dense in $L^p([0, 1]^{d_x}, \mathbb{R}^{d_y})$.

The lower bound $w_{\min} \geq \max\{d_x, d_y, 2\}$ in Theorem 1 follows from an existing lower bound $w_{\min} \geq \max\{d_x, d_y\}$ (Cai, 2023) and a lower bound $w_{\min} \geq 2$. Here, the intuition behind each lower bound $d_x, d_y$, and 2 is rather straightforward. If a network has width $d_x - 1$, then it must have the form $g(Mx)$ for some continuous function $g : \mathbb{R}^{d_x-1} \to \mathbb{R}^{d_y}$ and $M \in \mathbb{R}^{(d_x-1)\times d_x}$, which cannot universally approximate, e.g., consider approximating $\|x\|_2^2$. Likewise, if a network has width $d_y - 1$, then it must have the form $Nh(x)$ for some continuous function $h : \mathbb{R}^{d_x} \to \mathbb{R}^{d_y-1}$ and $N \in \mathbb{R}^{d_y \times (d_y-1)}$, which cannot universally approximate. Lastly, a RELU network of width 1 is monotone and hence, cannot approximate non-monotone functions. Combining these three arguments leads us to the lower bound $\max\{d_x, d_y, 2\}$ in Theorem 1. We note that our proof techniques are not restricted to RELU; they can be extended to various RELU-like activation functions.

**Theorem 2.** $w_{\min} = \max\{d_x, d_y, 2\}$ *for $\varphi$ networks to be dense in $L^p([0, 1]^{d_x}, \mathbb{R}^{d_y})$ if $\varphi \in \{$ELU, Leaky-RELU, SOFTPLUS, CELU, SELU$\}$, or $\varphi \in \{$GELU, SILU, MISH$\}$ and $d_x + d_y \geq 3$.*

Theorem 2 provides that for ELU, Leaky-RELU, SOFTPLUS, CELU, and SELU networks, width $\max\{d_x, d_y, 2\}$ is necessary and sufficient for universal approximation of $L^p([0, 1]^{d_x}, \mathbb{R}^{d_y})$. On the other hand, the minimum width of GELU, SILU, and MISH networks to be dense in $L^p([0, 1]^{d_x}, \mathbb{R}^{d_y})$ is $\max\{d_x, d_y, 2\}$ if $d_x + d_y \geq 3$. In particular, Theorem 2 can be further generalized to any continuous function $\rho$ such that RELU can be uniformly approximated by a $\rho$ network of width one on any compact domain, within an arbitrary uniform error. We present the proof for the upper bound $w_{\min} \leq \max\{d_x, d_y, 2\}$ in Theorem 1 in Section 4 while the proof for the matching lower bound in Theorem 1 and the proof of Theorem 2 are deferred to Appendix C and Appendix D.

We note that our proof techniques easily extend to RNNs and bidirectional RNNs: Theorems 24 and 25 in Appendix F shows that the same result in Theorems 1 and 2 also holds for RNNs. Furthermore, Theorem 26 in Appendix F shows that $w_{\min} \leq \max\{d_x, d_y, 2\}$ for bidirectional RNNs using any of RELU or RELU-LIKE activation functions to be dense in $L^p([0, 1]^{d_x}, \mathbb{R}^{d_y})$.

**Uniform approximation with general activation functions.** For RELU networks, it is known that $w_{\min}$ for $C([0, 1]^{d_x}, \mathbb{R}^{d_y})$ is greater than that for $L^p([0, 1]^{d_x}, \mathbb{R}^{d_y})$ in general. This is shown by the observation in (Park et al., 2021b): if $d_x = 1$ and $d_y = 2$, then width 2 is sufficient for RELU networks to be dense in $L^p([0, 1]^{d_x}, \mathbb{R}^{d_y})$, but insufficient to be dense in $C([0, 1]^{d_x}, \mathbb{R}^{d_y})$. However, whether this observation extends has been unknown. Our next theorem shows that a similar result holds for a wide class of activation functions and $d_x, d_y$.

**Theorem 3.** *For any continuous $\varphi : \mathbb{R} \to \mathbb{R}$ that can be uniformly approximated by a sequence of continuous injections, if $d_x < d_y \leq 2d_x$, then $w_{\min} \geq d_y + 1$ for $\varphi$ networks to be dense in $C([0, 1]^{d_x}, \mathbb{R}^{d_y})$.*

Theorem 3 states that $w_{\min} \geq d_y + 1$ for $C([0, 1]^{d_x}, \mathbb{R}^{d_y})$ if $d_y \in (d_x, 2d_x]$ and the activation function can be uniformly approximated by a sequence of continuous injections (e.g., any monotone continuous function such as RELU). This bound is tight for Leaky-RELU networks if $2d_x = d_y$, together with the matching upper bound $\max\{2d_x + 1, d_y\}$ on $w_{\min}$ (Hwang, 2023). Combined with Theorem 1, this result implies that $w_{\min}$ for RELU networks to be dense in $C([0, 1]^{d_x}, \mathbb{R}^{d_y})$ is strictly larger than that for $L^p([0, 1]^{d_x}, \mathbb{R}^{d_y})$ if $d_x < d_y \leq 2d_x$. With Theorem 2 and monotone RELU-LIKE activation functions, a similar observation can also be made.

We prove Theorem 3 by explicitly constructing a continuous target function that cannot be approximated by a $\varphi$ network of width $d_y$ in a small uniform distance where $\varphi$ is an activation function that can be uniformly approximated by a sequence of continuous one-to-one functions. In particular, based on topological arguments, we prove that any continuous function that uniformly approximates our target function within a small error has an intersection, i.e., it cannot be uniformly approximated by injective functions, which leads us to the statement of Theorem 3. We present a detailed proof of Theorem 3 including the formulation of our target function in Section 5.

We lastly note that all results with the domain $[0, 1]^{d_x}$ also hold for arbitrary compact domain $\mathcal{K} \subset \mathbb{R}^{d_x}$: if the target function $f^*$ is continuous, then one can always find $K > 0$ such that $\mathcal{K} \subset [-K, K]^{d_x}$ and continuously extend $f^*$ to $[-K, K]^{d_x}$ by the Tietze extension lemma (Munkres, 2000). If $f^*$ is $L^p$, then approximate $f^*$ by some continuous function and perform the extension.

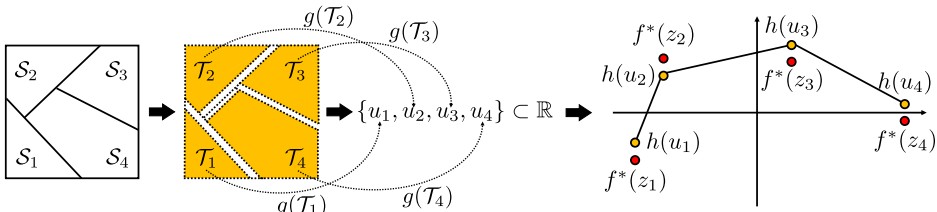

Figure 1: Illustration of our encoder and decoder when $d_x = 2$, $d_y = 1$ and $k = 4$. Our encoder $g$ first maps each element of $\{\mathcal{T}_1, \ldots, \mathcal{T}_4\}$ to distinct scalar codewords $u_1, \ldots, u_4$. Then, the decoder $h$ maps each codeword $u_i$ to $h(u_i) \approx f^*(z_i)$ for some $z_i \in \mathcal{T}_i$.

## 4    TIGHT UPPER BOUND ON MINIMUM WIDTH FOR $L^p$-APPROXIMATION

In this section, we prove the upper bound in Theorem 1 by explicitly constructing a RELU network of width $\max\{d_x, d_y, 2\}$ approximating a target function in $L^p([0, 1]^{d_x}, \mathbb{R}^{d_y})$. Since continuous functions on $[0, 1]^{d_x}$ are dense in $L^p([0, 1]^{d_x}, \mathbb{R}^{d_y})$ (Rudin, 1987), it suffices to prove the following lemma to show the upper bound. Here, we restrict the codomain to $[0, 1]^{d_y}$; however, this result can be easily extended to the codomain $\mathbb{R}^{d_y}$ since the range of $f^* \in C([0, 1]^{d_x}, \mathbb{R}^{d_y})$ is compact.

**Lemma 4.** *Let $\varepsilon > 0$, $p \geq 1$, and $f^* \in C([0, 1]^{d_x}, [0, 1]^{d_y})$. Then, there exists a RELU network $f : [0, 1]^{d_x} \to \mathbb{R}^{d_y}$ of width $\max\{d_x, d_y, 2\}$ such that $\|f - f^*\|_p \leq \varepsilon$.*

### 4.1    CODING SCHEME AND RELU NETWORK IMPLEMENTATION (PROOF OF LEMMA 4)

Our proof of Lemma 4 is based on the *coding scheme* that consists of two functions (Park et al., 2021b): the *encoder* and *decoder*. The *encoder* first transforms each input vector $x \in [0, 1]^{d_x}$ to a scalar-valued codeword containing the information of $x$; then the *decoder* maps each codeword to a target vector in $[0, 1]^{d_y}$ which approximates $f^*(x)$. Namely, the composition of these two functions approximates the target function. The precise operations of our encoder and decoder are as follows.

Suppose that a partition $\{\mathcal{S}_1, \ldots, \mathcal{S}_k\}$ of the domain $[0, 1]^{d_x}$ is given. Then, the encoder maps each input vector in $\mathcal{S}_i$ to some scalar-valued codeword $c_i$. Here, if the diameter of the set $\mathcal{S}_i$ is small enough, then it is reasonable to map vectors in $\mathcal{S}_i$ to the same codeword, say $c_i \in \mathbb{R}$, since the target function $f^*$ is uniformly continuous on $[0, 1]^{d_x}$, i.e., $f^*(x) \approx f^*(x')$ for all $x, x' \in \mathcal{S}_i$. However, since such an encoder is discontinuous in general, we approximate it using a RELU network via the following lemma. We present the main proof idea of Lemma 5 in Section 4.2 and defer the full proof to Appendix B.3.

**Lemma 5.** *For any $\alpha, \beta > 0$, there exist disjoint measurable sets $\mathcal{T}_1, \ldots, \mathcal{T}_k \subset [0, 1]^{d_x}$ and a RELU network $f : \mathbb{R}^{d_x} \to \mathbb{R}$ of width $\max\{d_x, 2\}$ such that*

- *$\mathrm{diam}(\mathcal{T}_i) \leq \alpha$ for all $i \in [k]$,*

- *$\mu_{d_x}\left(\bigcup_{i=1}^k \mathcal{T}_i\right) \geq 1 - \beta$, and*

- *$f(\mathcal{T}_i) = \{c_i\}$ for all $i \in [k]$, for some distinct $c_1, \ldots, c_k \in \mathbb{R}$.*

Lemma 5 states that there is an (approximate) encoder given by a RELU network $g$ of width $\max\{d_x, 2\}$ that can assign distinct codewords to $\mathcal{T}_1, \ldots, \mathcal{T}_k$. Here, $\mathcal{T}_1, \ldots, \mathcal{T}_k$ can be considered as an approximate partition since they are disjoint and cover at least $1 - \beta$ fraction of the domain for any $\beta > 0$. By choosing a small enough $\alpha$, we can have a small *information loss* of the input vectors in $\mathcal{T}_1, \ldots, \mathcal{T}_k$, incurred by encoding them via Lemma 5. We note that such an approximate encoder may map inputs that are not contained in $\mathcal{T}_1 \cup \cdots \cup \mathcal{T}_k$ to arbitrary values.

Once the encoder transforms all input vectors in $\mathcal{T}_i$ to a single codeword $c_i \in \mathbb{R}$, the decoder maps the codeword to a $d_y$-dimensional vector that approximates $f^*(\mathcal{T}_i)$. We implement the decoder using a RELU network using the following lemma, which is a corollary of Lemma 9 and Lemma 10 in Park et al. (2021b). See Appendix B.4 for its formal derivation.

**Lemma 6.** *For any $p \geq 1$, $\gamma > 0$, distinct $c_1, \ldots, c_k \in \mathbb{R}$, and $v_1, \ldots, v_k \in \mathbb{R}^{d_y}$, there exists a RELU network $f : \mathbb{R} \to [0, 1]^{d_y}$ of width $\max\{d_y, 2\}$ such that $\|f(c_i) - v_i\|_p \leq \gamma$ for all $i \in [k]$.*

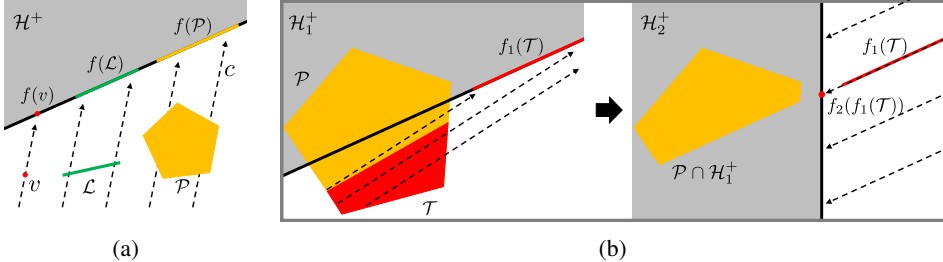

(a)                 (b)

Figure 2: Construction of $f$ in Lemma 7. (a) $f$ preserves points in the half-space $\mathcal{H}^+$ represented by the gray area and projects points outside of $\mathcal{H}^+$ to the boundary of $\mathcal{H}^+$. (b) Illustrations of mapping $\mathcal{T}$ to a single point disjoint to $\mathcal{P} \cap \mathcal{H}_1^+$ when $d_x = 2$: $f_1$ maps $\mathcal{T}$ onto the boundary of $\mathcal{H}_1^+$ and then $f_2$ maps $f_1(\mathcal{T})$ to the point $f_2(f_1(\mathcal{T}))$ while preserving points in $\mathcal{P} \cap \mathcal{H}_1^+$.

Lemma 4 follows from an (approximate) encoder and decoder in Lemmas 5 and 6. Let $\mathcal{T}_1, \ldots, \mathcal{T}_k$ be an approximate partition and $g$ be a ReLU network of width $\max\{d_x, 2\}$ in Lemma 5 with some $\alpha, \beta > 0$. Likewise, let $h$ be a ReLU network of width $\max\{d_y, 2\}$ in Lemma 6 with codewords $c_1, \ldots, c_k$ generated by $g$, $v_i = f^*(z_i)$ for some $z_i \in \mathcal{T}_i$ for all $i \in [k]$, and some $\gamma > 0$. Then, $f = h \circ g$ can be implemented by a ReLU network of width $\max\{d_x, d_y, 2\}$ and can approximate the target function $f^*$ in $\varepsilon$ error if we choose small enough $\alpha, \beta, \gamma$. Since the codomain of our decoder is $[0, 1]$, one can observe that $f(x) \in [0, 1]$ for any $x$ that is not contained in any of $\mathcal{T}_1, \ldots, \mathcal{T}_k$, i.e., they only incur a small $L^p$ error if $\beta$ is small enough. Figure 1 illustrates our encoder and decoder construction. See Appendix B.2 for our choices of $\alpha, \beta, \gamma$ achieving the statement of Lemma 4.

### 4.2    APPROXIMATING ENCODER USING ReLU NETWORK (PROOF SKETCH OF LEMMA 5)

In this section, we sketch the proof of Lemma 5 where the full proof is in Appendix B.3. To this end, we first introduce the following key lemma. The proof of Lemma 7 is deferred to Appendix B.5

**Lemma 7.** *For any* $d_x \in \mathbb{N}$*, a compact set* $\mathcal{K} \subset \mathbb{R}^{d_x}$*,* $a, c \in \mathbb{R}^{d_x}$ *such that* $a^\top c > 0$*, and* $b \in \mathbb{R}$*, there exists a two-layer* ReLU *network* $f : \mathcal{K} \to \mathbb{R}^n$ *of width* $d_x$ *such that*

$$
f(x) = \begin{cases} x & \text{if } a^\top x + b \geq 0 \\ x - \frac{a^\top x + b}{a^\top c} \times c & \text{if } a^\top x + b < 0 \end{cases}.
$$

Lemma 7 states that there exists a two-layer ReLU network of width $d_x$ on a compact domain that preserves the points in the half-space $\mathcal{H}^+ = \{x \in \mathbb{R}^{d_x} : a^\top x + b \geq 0\}$ and projects points not in $\mathcal{H}^+$ to the boundary of $\mathcal{H}^+$ along the direction determined by a vector $c$ as illustrated in Figure 2a.

This lemma has two important applications. First, for any bounded set, Lemma 7 enables us to project it onto a hyperplane (the boundary of $\mathcal{H}^+$), along a vector $c$. In other words, we can use Lemma 7 for decreasing a dimension of a bounded set or moving a point as illustrated in Figure 2a. Furthermore, given a polytope $\mathcal{P} \subset \mathbb{R}^{d_x}$ and a half-space $\mathcal{H}^+$ such that both $\mathcal{P} \cap \mathcal{H}^+$ and $\mathcal{P} \setminus \mathcal{H}^+$ are non-empty, we can construct a ReLU network of width $d_x$ that preserves points in $\mathcal{P} \cap \mathcal{H}^+$ and maps some $\mathcal{T} \subset \mathcal{P} \setminus \mathcal{H}^+$ with $\mu_{d_x}(\mathcal{T}) \approx \mu_{d_x}(\mathcal{P} \setminus \mathcal{H}^+)$ to a single point disjoint to $\mathcal{P} \cap \mathcal{H}^+$. As illustrated in Figure 2b, this can be done by mapping $\mathcal{P} \setminus \mathcal{H}^+$ onto the boundary of $\mathcal{H}^+$ first, and then, iteratively projecting a subset of the image of $\mathcal{P} \setminus \mathcal{H}^+$ (i.e., the image of $\mathcal{T}$) to a single point using Lemma 7. We note that the measure of $\mathcal{T}$ can be arbitrarily close to that of $\mathcal{P} \setminus \mathcal{H}^+$ by choosing a proper $c$ in Lemma 7 when projecting $\mathcal{P} \setminus \mathcal{H}^+$ onto the boundary of $\mathcal{H}^+$.

We now describe our construction of $f$ in Lemma 5. First, suppose that there is a partition $\{\mathcal{S}_1, \ldots, \mathcal{S}_k\}$ of the domain $[0, 1]^{d_x}$ where each $\mathcal{S}_i$ can be represented as

$$
\mathcal{S}_i = [0, 1]^{d_x} \cap \left( \bigcap_{j=1}^{i-1} \mathcal{H}_j^+ \right) \cap (\mathcal{H}_i^+)^c
$$

for some half-spaces $\mathcal{H}_1^+, \ldots, \mathcal{H}_k^+$; see the first image in Figure 3 for example. Suppose further that $\mathrm{diam}(\mathcal{S}_i) \leq \alpha$ for all $i \in [k]$. We note that such a partition always exists as stated in Lemma 10 in Appendix B.3. As in the second image of Figure 3, the most part of $\mathcal{S}_1$ ($\mathcal{T}_1$ in Figure 3) can be

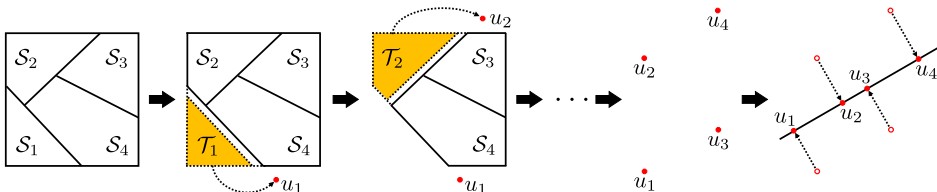

Figure 3: Illustration of the encoder when $d_x = 2$ and $k = 4$. For each partition $\mathcal{S}_i$, the encoder maps $\mathcal{T}_i \subset \mathcal{S}_i$ to some point $u_i \notin \{u_1, \ldots, u_{i-1}\} \cup \mathcal{S}_{i+1} \cup \cdots \cup \mathcal{S}_k$. After that, the encoder maps $u_1, \ldots, u_k$ to some distinct scalar values by projecting them.

mapped into a single point ($u_1$ in Figure 3) disjoint to $\mathcal{S}_2 \cup \cdots \cup \mathcal{S}_k$ using Lemma 7. Likewise, we map the most part of $\mathcal{S}_2$ ($\mathcal{T}_2$ in Figure 3) to a single point disjoint to $\{u_1\} \cup \mathcal{S}_3 \cup \cdots \cup \mathcal{S}_k$. Here, if $u_1 \notin \mathcal{H}_2^+$, we first move it so that $u_1 \in \mathcal{H}_2^+$ using Lemma 7 while preserving points in $\mathcal{S}_2 \cup \cdots \cup \mathcal{S}_k$. By repeating this procedure, we can consequently map the most parts ($\mathcal{T}_1, \ldots, \mathcal{T}_k$) of $\mathcal{S}_1, \ldots, \mathcal{S}_k$ to $k$ distinct points via a RELU network of width $d_x$ (the second last image in Figure 3). We finally project these points to distinct scalar values as illustrated in the last image in Figure 3. See Lemmas 11 and 12 in Appendix B.3 and its proof for the formal statements.

We note that our construction of a RELU network satisfies the three conditions in Lemma 5. The first condition is naturally satisfied since $\mathcal{T}_i \subset \mathcal{S}_i$ and $\mathrm{diam}(\mathcal{S}_i) \leq \alpha$. The second condition can also be satisfied since the measure of $\mathcal{T}_i$ can be arbitrarily close to that of $\mathcal{S}_i$ for all $i \in [k]$. Lastly, our construction maps each $\mathcal{T}_i$ to a distinct scalar value; this provides the third condition.

## 5 LOWER BOUND ON MINIMUM WIDTH FOR UNIFORM APPROXIMATION

In this section, we prove Theorem 3 by explicitly showing the existence of a continuous function $f^* : [0,1]^{d_x} \to \mathbb{R}^{d_y}$ that cannot be approximated by any $\varphi$ network of width $d_y$ within $1/3$ error in the uniform norm when $d_x < d_y \leq 2d_x$. Based on the following lemma, we assume that the activation function $\varphi$ is a continuous injection throughout the proof without loss of generality. The proof of Lemma 8 is presented in Appendix E.

**Lemma 8.** *Let $\sigma : \mathbb{R} \to \mathbb{R}$ be a continuous function that can be uniformly approximated by a sequence of continuous injections. Then, for any $\sigma$ network $f : [0,1]^{d_x} \to \mathbb{R}^{d_y}$ of width $w$ and for any $\varepsilon > 0$, there exists a $\varphi$ network $g : [0,1]^{d_x} \to \mathbb{R}^{d_y}$ of width $w$ such that $\varphi : \mathbb{R} \to \mathbb{R}$ is a continuous injection and $\|f - g\|_\infty \leq \varepsilon$.*

**Our choice of $f^*$.** We consider $f^*$ of the following form: for $r = d_y - d_x$, $x = (x_1, \ldots, x_{d_x}) \in [0,1]^{d_x}$, $\mathcal{D}_1 = [0, 1/3]^{d_x}$, and $\mathcal{D}_2 = [2/3, 1]^r \times \{1\}^{d_x - r}$,

$$f^*(x) = \begin{cases} (1 - 6x_1, 1 - 6x_2, \ldots, 1 - 6x_{d_x}, 0, \ldots, 0) & \text{if } x \in \mathcal{D}_1 \\ (0, \ldots, 0, 6x_1 - 5, 6x_2 - 5, \ldots, 6x_r - 5) & \text{if } x \in \mathcal{D}_2 \\ g^*(x) & \text{otherwise} \end{cases}$$

where $g^*$ is some continuous function that makes $f^*$ continuous; such $g^*$ always exists by the Tietze extension lemma and the pasting lemma (Munkres, 2000). We note that $f^*|_{\mathcal{D}_1}$ and $f^*|_{\mathcal{D}_2}$ are injections whose images are $[-1, 1]^{d_x} \times \{0\}^r$ and $\{0\}^{d_x} \times [-1, 1]^r$, respectively, i.e., $f^*(\mathcal{D}_1) \cap f^*(\mathcal{D}_2) = \{(0, \ldots, 0)\}$. See Figures 4a and 4b for illustrations of $\mathcal{D}_1, \mathcal{D}_2, f^*(\mathcal{D}_1)$, and $f^*(\mathcal{D}_2)$.

**Assumptions on $\varphi$ network approximating $f^*$.** Suppose for a contradiction that there are a continuous injection $\varphi : \mathbb{R} \to \mathbb{R}$ and a $\varphi$ network $f : [0,1]^{d_x} \to \mathbb{R}^{d_y}$ such that $\|f^* - f\|_\infty \leq 1/3$ and $f = t_L \circ \phi \circ \cdots \circ t_2 \circ \phi \circ t_1$ where $t_1 : \mathbb{R}^{d_x} \to \mathbb{R}^{d_y}$, $t_2, \ldots, t_L : \mathbb{R}^{d_y} \to \mathbb{R}^{d_y}$ are some affine transformations and $\phi : \mathbb{R}^{d_y} \to \mathbb{R}^{d_y}$ is a pointwise application of $\varphi$. Without loss of generality, we assume that $t_2, \ldots, t_L$ are invertible, as invertible affine transformations are dense in the space of affine transformations on bounded support, endowed with $\|\cdot\|_\infty$. Likewise, we assume that $t_1$ is injective. Since $\varphi$ is an injection, $f$ is also an injection, i.e.,

$$f(\mathcal{D}_1) \cap f(\mathcal{D}_2) = \emptyset. \tag{2}$$

However, one can expect that such $f$ cannot be injective as illustrated in Figure 4c. Based on this intuition, we now formally show a contradiction.

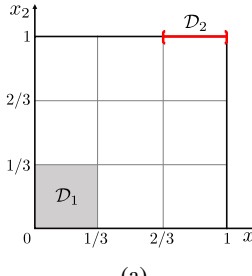 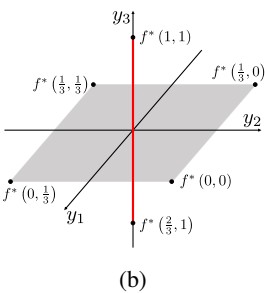 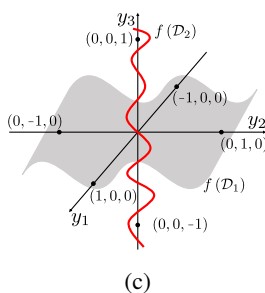

(a)                                    (b)                                    (c)

Figure 4: $\mathcal{D}_1, \mathcal{D}_2$ and their corresponding images of $f^*$ when $d_x = 2$ and $d_y = 3$ are illustrated by the grey squares and red lines in (a) and (b). One of the possible images of $f(\mathcal{D}_1)$ and $f(\mathcal{D}_2)$ are represented by the grey surface and red curve in (c).

**Proof by contradiction.** Define $h_1 : [-1, 1]^{d_x} \to \mathbb{R}^{d_y}$, $h_2 : [-1, 1]^r \to \mathbb{R}^{d_y}$, and $\psi : [-1, 1]^{d_y} \to [-1, 1]^{d_y}$ as follows: for $\alpha \in [-1, 1]^{d_x}$, $\beta \in [-1, 1]^r$, and $\gamma = (\alpha, \beta) \in [-1, 1]^{d_y}$,

$$h_1(\alpha) = f\big((\alpha + 1)/6\big), \quad h_2(\beta) = f\big((\beta + 5)/6, 1, \ldots, 1\big), \quad \psi(\gamma) = \frac{h_1(\alpha) - h_2(\beta)}{\|h_1(\alpha) - h_2(\beta)\|_\infty}.$$

From the definitions of $h_1, h_2$ and Eq. (2), one can observe that

$$h_1([-1, 1]^{d_x}) \cap h_2([-1, 1]^r) = f(\mathcal{D}_1) \cap f(\mathcal{D}_2) = \emptyset,$$

i.e., $\psi$ is well-defined (and continuous) as $\|h_1(\alpha) - h_2(\beta)\|_\infty > 0$ for all $(\alpha, \beta) \in [-1, 1]^{d_y}$. From the definitions of $\psi$ and the infinity norm, it holds that

*(i)* $\psi([-1, 1]^{d_y}) \subset [-1, 1]^{d_y}$ and

*(ii)* for each $\gamma \in [-1, 1]^{d_y}$, there exists $i \in [d_y]$ such that $\psi(\gamma)_i \in \{-1, 1\}$.

By *(i)*, continuity of $\psi$, and the Brouwer's fixed point theorem (Lemma 9), there exists $\gamma^* = (\alpha^*, \beta^*) \in [-1, 1]^{d_y}$ such that $\psi(\gamma^*) = \gamma^*$. Furthermore, by *(ii)*, there should be $i^* \in [d_y]$ such that $\gamma^*_{i^*} \in \{-1, 1\}$. However, we now show that such $i^*$ does not exist.

Suppose that there is $i^* \in [d_x]$ satisfying $\gamma^*_{i^*} = \psi(\gamma^*)_{i^*} = 1$. Then, by the definitions of $f, h_1, h_2$ and the assumption $\|f^* - f\|_\infty \le 1/3$, the following holds for all $z = (x, y) \in [-1, 1]^{d_y}$ with $z_{i^*} = 1$: $h_1(x)_{i^*} \in [-4/3, -2/3]$ and $h_2(y)_{i^*} \in [-1/3, 1/3]$. This implies that $h_1(x)_{i^*} - h_2(y)_{i^*}$ is negative for all $(x, y) \in [-1, 1]^{d_y}$, and therefore, $\psi(\gamma^*)_{i^*} < 0$ which contradicts $\psi(\gamma^*)_{i^*} = 1$. Using similar arguments, one can also show the contradiction for the two remaining cases: $i^* \in [d_x]$ and $\gamma^*_{i^*} = -1$; and $i^* \in [d_y] \setminus [d_x]$ and $\gamma^*_{i^*} \in \{-1, 1\}$. In other words, $\gamma^*_i$ cannot be any of $\{-1, 1\}$ for all $i \in [d_y]$. This contradicts *(ii)* and proves Theorem 3.

**Lemma 9** (Brouwer's fixed-point theorem (Florenzano, 2003)). *For any non-empty compact convex set $\mathcal{K} \subset \mathbb{R}^n$ and continuous function $f : \mathcal{K} \to \mathcal{K}$, there exists $x^* \in \mathcal{K}$ such that $f(x^*) = x^*$.*

## 6 CONCLUSION

Identifying the universal approximation property of deep neural networks is a fundamental problem in the theory of deep learning. Several works have tried to characterize the minimum width enabling universal approximation; however, only a few of them succeed in finding the exact minimum width. In this work, we first prove that the minimum width of networks using RELU or RELU-LIKE activation functions is $\max\{d_x, d_y, 2\}$ for universal approximation in $L^p([0, 1]^{d_x}, \mathbb{R}^{d_y})$. Compared to the existing result that width $\max\{d_x + 1, d_y\}$ is necessary and sufficient for RELU networks to be dense in $L^p(\mathbb{R}^{d_x}, \mathbb{R}^{d_y})$, our result shows a dichotomy between universal approximation on a compact domain and the whole Euclidean space. Furthermore, using a topological argument, we improve the lower bound on the minimum width for uniform approximation when the activation function can be uniformly approximated by a sequence of continuous one-to-one functions: the minimum width is at least $d_y + 1$ if $d_x < d_y \le 2d_x$, which is shown to be tight for Leaky-RELU networks if $2d_x = d_y$. This generalizes prior results showing a gap between $L^p$ and uniform approximations to general activation functions and input/output dimensions. We believe that our results and proof techniques can help better understand the expressive power of deep neural networks.

ACKNOWLEDGEMENTS

NK and SP were supported by Institute of Information & communications Technology Planning & Evaluation (IITP) grant funded by the Korea government (MSIT) (No. 2019-0-00079, Artificial Intelligence Graduate School Program, Korea University) and Basic Science Research Program through the National Research Foundation of Korea (NRF) funded by the Ministry of Education (2022R1F1A1076180).

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

# A  DEFINITION OF ACTIVATION FUNCTIONS

In this section, we introduce the definitions of activation functions that we mainly focus on.

- ReLU (RELU):

$$\mathrm{RELU}(x) = \begin{cases} x & \text{if } x > 0 \\ 0 & \text{if } x \leq 0 \end{cases}.$$

- Softplus (SOFTPLUS): for $\alpha > 0$,

$$\mathrm{SOFTPLUS}(x; \alpha) = \frac{1}{\alpha} \log(1 + \exp(\alpha x)).$$

- LeakyReLU (Leaky-RELU): for $\alpha \in (0, 1)$,

$$\text{Leaky-RELU}(x; \alpha) = \begin{cases} x & \text{if } x > 0 \\ \alpha x & \text{if } x \leq 0 \end{cases}.$$

- Exponential Linear Unit (ELU): for $\alpha > 0$,

$$\mathrm{ELU}(x; \alpha) = \begin{cases} x & \text{if } x > 0 \\ \alpha \left(\exp(x) - 1\right) & \text{if } x \leq 0 \end{cases}.$$

- Continuously differentiable Exponential Linear Unit (CELU): for $\alpha > 0$,

$$\mathrm{CELU}(x; \alpha) = \begin{cases} x & \text{if } x > 0 \\ \alpha \left(\exp(x/\alpha) - 1\right) & \text{if } x \leq 0 \end{cases}.$$

- Scaled Exponential Linear Unit (SELU): for $\lambda > 1$ and $\alpha > 0$,

$$\mathrm{SELU}(x; \lambda, \alpha) = \lambda \times \begin{cases} x & \text{if } x > 0 \\ \alpha \left(\exp(x) - 1\right) & \text{if } x \leq 0 \end{cases}.$$

- Gaussian Error Linear Unit (GELU):

$$\mathrm{GELU}(x) = x \times \Phi(x)$$

  where $\Phi(x)$ is the cumulative distribution function of the standard normal distribution.

- Sigmoid Linear Unit (SILU):

$$\mathrm{SILU}(x) = x \times \mathrm{SIGMOID}(x)$$

  where $\mathrm{SIGMOID}(x) = 1/\left(1 + \exp(-x)\right)$ is the sigmoid activation function.

- Mish (MISH):

$$\mathrm{MISH}(x) = x \times \mathrm{TANH}(\mathrm{SOFTPLUS}(x; 1))$$

  where $\mathrm{TANH}(x) = (\exp(x) - \exp(-x))/(\exp(x) + \exp(-x))$.

## B  PROOF OF UPPER BOUND IN THEOREM 1

### B.1  ADDITIONAL NOTATIONS

Prior to delving into the proof of Theorem 1, we first introduce some notations that will be frequently employed in the subsequent sections. Given a set $\mathcal{S} \subset \mathbb{R}^n$, $int(\mathcal{S})$ denotes the interior of $\mathcal{S}$, and $bd(\mathcal{S})$ denotes the boundary of $\mathcal{S}$. For $(a, b) \in \mathbb{R}^n \times \mathbb{R}$, $\mathcal{H}(a, b) \triangleq \{x \in \mathbb{R}^n : a^\top x + b = 0\}$ denotes the hyperplane parameterized by $a$ and $b$. Likewise, we use $\mathcal{H}^+(a, b) \triangleq \{x \in \mathbb{R}^n : a^\top x + b \geq 0\}$ and $\mathcal{H}^-(a, b) \triangleq \{x \in \mathbb{R}^n : a^\top x + b \leq 0\}$ for denoting corresponding upper half-space and lower half-space, respectively.

### B.2  OUR CHOICES OF $\alpha, \beta, \gamma$

We use a small enough $\alpha > 0$ so that $\omega_{p,2,f^*}(\alpha) \leq \varepsilon/2^{1+1/p}$, $\beta = \varepsilon^p/(2d_y)$, and $\gamma = \varepsilon/2^{1+1/p}$. Here, $\omega_{p,2,f^*}$ denotes the modulus of continuity of $f^*$ in the $p$-norm and 2-norm: $\|f^*(x) - f^*(x')\|_p \leq \omega_{p,2,f^*}(\|x - x'\|_2)$ for all $x, x' \in [0,1]^{d_x}$. We note that such $\omega_{p,2,f^*}$ is well-defined on $[0,1]^{d_x}$ since $f^*$ is uniformly continuous on $[0,1]^{d_x}$ (continuous function on a compact set). Namely, such $\alpha$ always exists for all $\varepsilon > 0$. Then, we have

$$
\begin{aligned}
\|f^* - f\|_p^p &= \int_{[0,1]^{d_x}} \|f^*(x) - f(x)\|_p^p d\mu_{d_x} \\
&= \int_{[0,1]^{d_x} \setminus \bigcup_{i=1}^k \mathcal{T}_i} \|f^*(x) - f(x)\|_p^p d\mu_{d_x} + \int_{\bigcup_{i=1}^k \mathcal{T}_i} \|f^*(x) - f(x)\|_p^p d\mu_{d_x} \\
&\leq d_y \times \mu_{d_x} \left( [0,1]^{d_x} \setminus \bigcup_{i=1}^k \mathcal{T}_i \right) + \sum_{i=1}^k \int_{\mathcal{T}_i} (\|f(x) - f^*(x)\|_p)^p d\mu_{d_x} \\
&\leq d_y \times \beta + \sum_{i=1}^k \int_{\mathcal{T}_i} (\|f^*(x) - f^*(z_i)\|_p + \|f(x) - f^*(z_i)\|_p)^p d\mu_{d_x} \\
&\leq d_y \times \beta + \sum_{i=1}^k \int_{\mathcal{T}_i} (\omega_{p,2,f^*}(\alpha) + \gamma)^p d\mu_{d_x} \\
&\leq d_y \times \beta + (\omega_{p,2,f^*}(\alpha) + \gamma)^p \leq \varepsilon^p
\end{aligned}
$$

where $z_i \in \mathcal{T}_i$ for all $i$.

This leads us to the statement of Lemma 4.

### B.3  PROOF OF LEMMA 5

We prove Lemma 5 in this section. To this end, we introduce the following lemma. The proof of Lemma 10 is presented in Appendix B.6. Here, $\mathcal{H}^+(a, b)$ is defined in Appendix B.1

**Lemma 10.** *Let* $\mathcal{K}_0 = [0,1]^n$. *For any* $\delta > 0$, *there exist* $m \in \mathbb{N}$ *and* $(a_1, b_1), \ldots, (a_m, b_m) \in \mathbb{R}^n \times \mathbb{R}$ *such that for* $\mathcal{K}_i = \mathcal{K}_0 \cap \mathcal{H}_i^+$ *where* $\mathcal{H}_i^+ = \bigcap_{j=1}^i \mathcal{H}^+(a_j, b_j)$ *for all* $i \in [m]$,

$$\mathsf{diam}(\mathcal{K}_0 \setminus \mathcal{K}_1), \mathsf{diam}(\mathcal{K}_1 \setminus \mathcal{K}_2), \ldots, \mathsf{diam}(\mathcal{K}_{m-1} \setminus \mathcal{K}_m) \leq \delta \quad and \quad \mathcal{K}_m = \emptyset.$$

Lemma 10 ensures the existence of the partition $\{\mathcal{S}_1, \ldots, \mathcal{S}_k\}$ of the domain $[0,1]^{d_x}$ such that the diameter of each $\mathcal{S}_i$ is upper bounded by given $\alpha > 0$ where each $\mathcal{S}_i$ can be represented as

$$\mathcal{S}_i = [0,1]^{d_x} \cap \left( \bigcap_{j=1}^{i-1} \mathcal{H}^+(a_j, b_j) \right) \cap \left( \mathcal{H}^+(a_i, b_i) \right)^c$$

for some $(a_1, b_1), \ldots, (a_i, b_i) \in \mathbb{R}^n \times \mathbb{R}$.

To show the existence of an approximation of a partition $\{\mathcal{T}_1, \ldots, \mathcal{T}_k\}$ and a RELU network of width $\max\{d_x, 2\}$ that maps $\mathcal{T}_1, \ldots, \mathcal{T}_k$ to some distinct points, we introduce the following lemma. The proof of Lemma 11 is presented in Appendix B.7.

**Lemma 11.** *Let $n \in \mathbb{N}$, $\mathcal{P} \subset \mathbb{R}^n$ be a convex polytope and $u_1, \ldots, u_k \in \mathbb{R}^n \setminus \mathcal{P}$ be distinct points. Let $\mathcal{H}^+ \subset \mathbb{R}^n$ be a closed half-space such that $\mathcal{P} \cap \mathcal{H}^+ \neq \emptyset$ and $\mathcal{P} \setminus \mathcal{H}^+ \neq \emptyset$. Let $m = \max\{n, 2\}$. Then for any $\delta \in (0, 1)$, there exists a $\mathrm{RELU}$ network $f : \mathbb{R}^n \to \mathbb{R}^m$ of width $m$ satisfying the following properties:*

- *$f(x) = x$ for all $x \in \mathcal{P} \cap \mathcal{H}^+$,*

- *there exist distinct $v_1, \ldots, v_k \in \mathbb{R}^m \setminus (\mathcal{P} \cap \mathcal{H}^+)$ such that $f(u_i) = v_i$ for all $i \in [k]$, and*

- *there exist $\mathcal{S} \subset \mathcal{P} \setminus \mathcal{H}^+$ and $v_{k+1} \in \mathbb{R}^m \setminus ((\mathcal{P} \cap \mathcal{H}^+) \cup \{v_1, \ldots, v_k\})$ such that $\mu_n(\mathcal{S}) \geq \delta \cdot \mu_n(\mathcal{P} \setminus \mathcal{H}^+)$ and $f(\mathcal{S}) = \{v_{k+1}\}$.*

Lemma 11 indicates that for each $\mathcal{S}_i$, there is a $\mathrm{RELU}$ network $g_i$ of width $\max\{d_x, 2\}$ such that $g_i$ (i) preserves the points in $\mathcal{S}_{i+1} \cup \cdots \cup \mathcal{S}_k$, (ii) transfers the $u_1, \ldots, u_{i-1}$ (containing "information" that we want to preserve) to some distinct points $v_1, \ldots, v_{i-1}$ not contained in $\mathcal{S}_{i+1} \cup \cdots \cup \mathcal{S}_k$, and (iii) embeds the most part of $\mathcal{S}_i$ (i.e. $\mathcal{T}_i$) to the point $v_i$ distinct to existing points $v_1, \ldots, v_{i-1}$ and also not contained in $\mathcal{S}_{i+1} \cup \cdots \cup \mathcal{S}_k$. Hence, by repeatedly applying Lemma 11, we can find $\{\mathcal{T}_1, \ldots, \mathcal{T}_k\}$ and a $\mathrm{RELU}$ network $g$ of width $\max\{d_x, 2\}$ such that $\mu_n(\mathcal{T}_i) \geq \mu_n(\mathcal{S}_i) - \beta/k$ for all $i \in [k]$ and the network maps each $\mathcal{T}_i$ to $u_i \in \mathbb{R}^m$ for some distinct $u_1, \ldots, u_k$ where $m = \max\{d_x, 2\}$.

Lastly, we introduce the following lemma, which demonstrates the existence of a projection map that maps finite distinct points to some distinct scalar values. The proof of Lemma 12 is presented in Appendix B.8.

**Lemma 12.** *For any $n \geq 2$ and distinct $v_1, \ldots, v_k \in \mathbb{R}^n$, there exists $a \in \mathbb{R}^n$ such that $a^\top v_1, \ldots, a^\top v_k$ are also distinct.*

By Lemma 12, there exists an affine map $h$ that maps $u_1, \ldots, u_k$ to distinct scalar values. Then, choosing $f = h \circ g$ completes the proof of Lemma 5.

### B.4 PROOF OF LEMMA 6

In this section, we prove Lemma 6 by explicitly constructing the target $f$. As aforementioned, the proof of Lemma 6 is a corollary of Lemma 9 and Lemma 10 in (Park et al., 2021b).

Before describing our proof details, we first introduce some functions introduced in (Park et al., 2021b). A quantization function $q_n : [0, 1] \to \mathcal{C}_n$ for $n \in \mathbb{N}$ and $\mathcal{C}_n \triangleq \{0, 2^{-n}, 2 \times 2^{-n}, 3 \times 2^{-n}, \ldots, 1 - 2^{-n}\}$ is defined as

$$q_n(x) = \max\{c \in \mathcal{C}_n : c \leq x\},$$

an encoder $\mathrm{encode}_K : \mathbb{R}^{d_x} \to \mathcal{C}_{d_x K}$ for some $K \in \mathbb{N}$ is defined as

$$\mathrm{encode}_K(x) = \sum_{i=1}^{d_x} q_K(x_i) \times 2^{-(i-1)K},$$

and a decoder $\mathrm{decode}_M : \mathcal{C}_{d_y M} \to \mathcal{C}_M^{d_y}$ is defined as

$$\mathrm{decode}_M(c) = \hat{x} \quad \text{where} \quad \{\hat{x}\} = \mathrm{encode}_M^{-1}(c) \cap \mathcal{C}_M^{d_y}.$$

Namely, $\mathrm{encode}_K$ quantizes every coordinate of input up to $K$-bits and then concatenates whole coordinates into a one-dimensional scalar value. And, $\mathrm{decode}_M$ decodes one-dimensional codewords to $d_y$-dimensional codewords.

Now, we introduce the following lemmas presented in (Park et al., 2021b).

**Lemma 13** (Lemma 9 in (Park et al., 2021b)). *For any $m \in \mathbb{N}$, $0 \leq \alpha_1 < \alpha_2 < \cdots < \alpha_m \leq 1$, and $\beta_1, \ldots, \beta_m \in \mathbb{R}$, there exists a $\mathrm{RELU}$ network $f : [0, 1] \to \mathbb{R}$ of width 2 such that*

$$f(\alpha_i) = \beta_i \quad \text{for all } i \in [m].$$

**Lemma 14** (Lemma 10 in (Park et al., 2021b)). *For any $d_y, M \in \mathbb{N}$, there exists a $\mathrm{RELU}$ network $f : \mathbb{R} \to \mathbb{R}^{d_y}$ of width $d_y$ such that for any $c \in \mathcal{C}_{d_y M}$,*

$$f(c) = (b_1, \ldots, b_{d_y})$$

*where $b_1, \ldots, b_{d_y} \in \mathcal{C}_M$ satisfying $c = \sum_{i=1}^{d_y} b_i \times 2^{-(i-1)M}$. Furthermore, it holds that $f(\mathbb{R}) \subset [0,1]^{d_y}$.*

Using Lemma 13, we first exactly construct a RELU network $g$ of width 2 which maps each code-word $c_i$ to the corresponding encoded target vector $\mathrm{encode}_M(v_i) \in \mathcal{C}_{d_y M}$ for some $M \in \mathbb{N}$; we will assign an explicit value to $M$ later. Next, by Lemma 14, we explicitly construct a RELU network $h$ of width $d_y$ which maps each $\mathrm{encode}_M(v_i)$ to being the $d_y$-dimensional quantized target vector $v_i^\dagger$ where $v_i^\dagger = (v_{i,1}^\dagger, \ldots, v_{i,d_y}^\dagger)$ and $v_{i,j}^\dagger = q_M(v_{i,j})$ for each $j \in [d_y]$.

Let $f$ be the composition of RELU networks $g$ and $h$. That is, $f$ is a RELU network of width $\max\{d_y, 2\}$. From the construction of $g$ and $h$, the error between $f(c_i) = v_i^\dagger$ and $v_i$ is only incurred from the quantization process. Hence, choosing sufficiently large $M \in \mathbb{N}$ such that $d_y^{1/p} \times 2^{-M} \leq \gamma$ completes the proof of Lemma 6.

For the sake of completeness, we provide proofs of Lemma 13 and Lemma 14, which are from (Park et al., 2021b).

*Proof of Lemma 13.* Consider the following piecewise linear function $f^* : [0,1] \to \mathbb{R}$ with $m+1$ pieces which satisfies the statement of Lemma 13:

$$f(x) = \begin{cases} \beta_1 & \text{if } x \in [0, \alpha_1) \\ \beta_i + \dfrac{\beta_{i+1} - \beta_i}{\alpha_{i+1} - \alpha_i}(x - \alpha_i) & \text{if } x \in [\alpha_i, \alpha_{i+1}) \text{ for some } i \in [m-1] \ . \\ \beta_m & \text{if } x \in [\alpha_m, 1] \end{cases}$$

From Lemma 15, we can construct a RELU network $f : [0,1] \to \mathbb{R}$ of width 2 satisfying $f^*(x) = f(x)$ for all $x \in [0,1]$. This completes the proof of Lemma 13. $\square$

**Lemma 15** (Lemma 14 in (Park et al., 2021b)). *For any compact interval $\mathcal{I} \subset \mathbb{R}$, for any continuous piecewise linear function $f^* : \mathcal{I} \subset \mathbb{R}$ with $P$ linear pieces, there exists a RELU network $f$ of width 2 such that $f(x) = f^*(x)$ for all $x \in \mathcal{I}$.*

*Proof of Lemma 15.* Suppose that $f^*$ is linear on $P$ pieces $[\min \mathcal{I}, x_1), [x_1, x_2), \ldots, [x_{P-1}, \max \mathcal{I}]$ and defined as

$$f(x) = \begin{cases} a_1 \times x + b_1 & \text{if } x \in [\min \mathcal{I}, x_1) \\ a_2 \times x + b_2 & \text{if } x \in [x_1, x_2) \\ \quad \vdots \\ a_P \times x + b_P & \text{if } x \in [x_{P-1}, \max \mathcal{I}] \end{cases}$$

for some $a_i, b_i \in \mathbb{R}$ satisfying $a_i \times x_i + b_i = a_{i+1} \times x_i + b_{i+1}$. Without loss of generality, we assume that $\min \mathcal{I} = 0$.

Now, we prove that for any $P \geq 1$, there exists a RELU network $f : \mathcal{I} \to \mathbb{R}^2$ of width 2 such that $f(x)_1 = \mathrm{RELU}(x - x_{P-1})$ and $f(x)_2 = f^*(x)$. Then, the RELU network $f(x)_2$ completes the proof. We use the mathematical induction for $P$ to prove the existence of corresponding $f$. When $P = 1$, choosing $f(x)_1 = \mathrm{RELU}(x)$ and $f(x)_2 = a_1 \times \mathrm{RELU}(x) + b_1$ satisfies the desired property. Here, consider $P > 1$. Then, from the induction hypothesis, there exists a RELU network $g$ of width 2 such that

$$g(x)_1 = \mathrm{RELU}(x - x_{P-2})$$

$$g(x)_2 = \begin{cases} a_1 \times x + b_1 & \text{if } x \in [\min \mathcal{I}, x_1) \\ a_2 \times x + b_2 & \text{if } x \in [x_1, x_2) \\ \quad \vdots \\ a_{P-1} \times x + b_{P-1} & \text{if } x \in [x_{P-2}, \max \mathcal{I}] \end{cases}$$

Then, the following construction of $f$ completes the proof of the mathematical induction:

$$f(x) = h_2 \circ \phi \circ h_1 \circ g(x)$$
$$h_1(x, z) = (x - x_{P-1} + x_{P-2}, z - K)$$
$$\phi(x, z) = (\text{RELU}(x), \text{RELU}(z))$$
$$h_2(x, z) = (x, K + z + (a_P - a_{P-1}) \times x)$$

where $K = \min_i \min_{x \in \mathcal{I}} \{a_i \times x + b_i\}$. Hence, this completes the proof of Lemma 15. $\qquad\square$

*Proof of Lemma 14.* We first introduce the following lemma.

**Lemma 16** (Lemma 15 in (Park et al., 2021b))**.** *For any $M \in \mathbb{N}$, for any $\delta > 0$, there exists a* RELU *network of $f : \mathbb{R} \to \mathbb{R}^2$ of width 2 such that for all $x \in [0, 1] \setminus \mathcal{D}_{M,\delta}$,*

$$f(x) = (y_1(x), y_2(x)), \quad where \quad y_1(x) = q_M(x), \quad y_2(x) = 2^M \times (x - q_M(x)), \qquad (3)$$

*and $\mathcal{D}_{M,\delta} = \bigcup_{i=1}^{2^M - 1} (i \times 2^{-M} - \delta, i \times 2^{-M})$. Furthermore, it holds that*

$$f(\mathbb{R}) \subset [0, 1 - 2^{-M}] \times [0, 1]. \qquad (4)$$

Fix some $\delta < 2^{-d_y M}$. Then, Lemma 16 indicates that there exists a RELU network $g$ of width 2 satisfying (3) on $\mathcal{C}_{d_y M}$ and (4) since $C_{d_y M} \subset [0, 1] \setminus \mathcal{D}_{M,\delta}$. Such $g$ enables us to extract the first $M$ bits of the binary representation of $c \in \mathcal{C}_{d_y M}$: $g(c)_1$ is the first coordinate of $\text{decode}_M(c)$ while $g(c)_2 \in C_{d_y-1 M}$ contains remaining information about other coordinates of $\text{decode}_M(c)$. Therefore, if we iteratively apply $g$ to the second output of the previous composition of $g$ and pass through all first outputs of the previous compositions of $g$, then we finally recover whole coordinates of $\text{decode}_M(c)$ within $d_y - 1$ compositions of $g$. Our construction of $f$ is such iterative $d_y - 1$ compositions of $g$ which can be implemented by a RELU network of width $d_y$. Moreover, (4) in Lemma 16 allows us to achieve $f(\mathbb{R}) \subset [0, 1]^{d_y}$. This completes the proof of Lemma 14. $\qquad\square$

*Proof of Lemma 16.* First of all, we clip the input to be in $[0, 1]$ using the following RELU network of width 1.

$$\min\{\max\{x, 0\}, 1\} = 1 - \text{RELU}(1 - \text{RELU}(x))$$

Then, we apply $g_\ell : [0, 1] \to [0, 1]^2$ defined as

$$g_\ell(x)_1 = x$$

$$g_\ell(x)_2 = \begin{cases} 0 & \text{if } x \in [0, 2^{-M} - \delta] \\ \delta^{-1} 2^{-M} \times (x - 2^{-M} + \delta) & \text{if } x \in (2^{-M} - \delta, 2^{-M}) \\ 2^{-M} & \text{if } x \in [2^{-M}, 2 \times 2^{-M} - \delta] \\ \delta^{-1} 2^{-M} \times (x - 2 \times 2^{-M} + \delta) + 2^{-M} & \text{if } x \in (2 \times 2^{-M} - \delta, 2 \times 2^{-M}) \\ \quad \vdots \\ (\ell - 1) \times 2^{-M} & \text{if } x \in [(\ell - 1) \times 2^{-M}, 1] \end{cases}$$

From the definition of $g_\ell$, one can observe that $g_{2^M}(x)_2 = q_M(x)$ for $x \in [0, 1] \setminus \mathcal{D}_{M,\delta}$. Therefore, once we implement $g_{2^M}(x)$ using a RELU network $g$ of width 2, and then constructing $f$ as

$$f(x) = (g(z)_2, 2^M \times (g(z)_1 - g(z)_2))$$
$$z = \min\{\max\{x, 0\}, 1\} = 1 - \text{RELU}(1 - \text{RELU}(x))$$

completes the proof of Lemma 16. Now, we construct a RELU network $g$ of width 2 which implements $g_{2^M}$. One can observe that $g_1(x)_2 = 0$ and

$$g_{\ell+1}(x)_2 = \min \left\{ \ell \times 2^{-M}, \max\{\delta^{-1} 2^{-M} \times (x - \ell \times 2^{-M} + \delta) + (\ell - 1) \times 2^{-M}, g_\ell(x)_2\} \right\}$$

for all $x \in [0, 1]$. To this end, we introduce the following definition and lemma.

**Definition 1** (Definition 1 in (Hanin and Sellke, 2017)). $f : \mathbb{R}^{d_x} \to \mathbb{R}^{d_y}$ *is a max-min string of length $L \geq 1$ if there exist affine transformations $h_1, \dots, h_L$ such that*

$$h(x) = \tau_{L-1}(h_L(x), \tau_{L-2}(h_{L-1}(x), \dots, \tau_2(h_3(x), \tau_1(h_2(x), h_1(x))) \dots),$$

*where each $\tau_\ell$ is either a coordinate-wise $\max\{\cdot, \cdot\}$ or $\min\{\cdot, \cdot\}$.*

**Lemma 17** (Proposition 2 in (Hanin and Sellke, 2017)). *For any max-min string $f : \mathbb{R}^{d_x} \to \mathbb{R}^{d_y}$ of length $L$, for any compact $\mathcal{K} \subset \mathbb{R}^{d_x}$, there exists a RELU network $g : \mathbb{R}^{d_x} \to \mathbb{R}^{d_x} \times \mathbb{R}^{d_y}$ of $L$ layers and width $d_x + d_y$ such that for all $x \in \mathcal{K}$,*

$$g(x) = (y_1(x), y_2(x)), \quad \text{where} \quad y_1(x) = x \quad \text{and} \quad y_2(x) = f(x).$$

Notably, $g_{2^M}(x)$ is a max-min string so that there exists a RELU network $g$ of width 2 satisfying $g(x)_2 = g_{2^M}(x) = q_M(x)$ for all $x \in [0,1] \setminus \mathcal{D}_{M,\delta}$. This completes the proof of Lemma 16. $\qquad\square$

## B.5 PROOF OF LEMMA 7

Let $a_1 = a$, $\{a_2, \dots, a_n\}$ be a basis of the hyperplane $\{x \in \mathbb{R}^n : c^\top x = 0\}$, and let $A = [a_1, \dots, a_n]^\top \in \mathbb{R}^{n \times n}$. Since $a^\top c \neq 0$, $A$ is invertible. Choose

$$K = -1 \times \min_{i \in \{2, \dots, n\}} \inf_{x \in \mathcal{K}} a_i^\top x$$

and $v = (b, K, \dots, K) \in \mathbb{R}^n$. Then we claim that choosing

$$f(x) = A^{-1}(\text{RELU}(Ax + v) - v)$$

completes the proof. Here, we use RELU for multi-dimensional input by applying RELU element-wise. From our choice of $A$, $v$, and $K$, if $a_1^\top x + b \geq 0$, then $f(x) = x$. Suppose that $a_1^\top x + b < 0$. In this case, from our choice of $K$, the second to the last coordinates of $\text{RELU}(Ax + v)$ are identical to that of $Ax + v$. Namely, we have

$$f(x) = A^{-1}\left((Ax + v - (a_1^\top x + b)e_1) - v\right) = x - (a_1^\top x + b)A^{-1}e_1 \tag{5}$$

where $e_1 = (1, 0, \dots, 0) \in \mathbb{R}^n$. Since $a_1^\top c > 0$ and $a_i^\top c = 0$ for all $i \in \{2, \dots, n\}$, the first column of $A^{-1}$ (i.e., $A^{-1}e_1$) must be $c/a_1^\top c = c/a^\top c$. Therefore, by Eq. (5), it holds that

$$f(x) = x - \frac{a^\top x + b}{a^\top c} \times c.$$

This completes the proof of Lemma 7.

## B.6 PROOF OF LEMMA 10

The statement of Lemma 10 directly follows from repeatedly applying Claim 18. Specifically, we iteratively construct $(a_1, b_1), \dots$ and corresponding $\mathcal{K}_1, \dots$ using Claim 18 so that $\text{diam}(\mathcal{K}_0 \setminus \mathcal{K}_1), \dots \leq \delta$ until $\text{diam}(\mathcal{K}_r) \leq \delta$ for some $r \in \mathbb{N}$. Then, we choose $(a_{r+1}, b_{r+1})$ such that

$$\mathcal{K}_{r+1} = \mathcal{K}_r \cap \mathcal{H}^+(a_{r+1}, b_{r+1}) = \emptyset.$$

Setting $m = r + 1$ completes the proof.

**Claim 18.** *For any $\delta > 0$ and bounded set $\mathcal{R}_0 \subset \mathbb{R}^n$ with $\text{diam}(\mathcal{R}_0) \leq D$ for some $D > 0$, there exist $k \in \mathbb{N}$ and $(c_1, d_1), \dots, (c_k, d_k) \subset \mathbb{R}^n \times \mathbb{R}$ such that*

- $\text{diam}(\mathcal{R}_0 \setminus \mathcal{R}_1), \dots, \text{diam}(\mathcal{R}_{k-1} \setminus \mathcal{R}_k) \leq \delta$ *and*

- $\text{diam}(\mathcal{R}_k) \leq \max\{D - \delta^2/(4D), 0\}$

*where $\mathcal{R}_i = \mathcal{R}_0 \cap \left(\bigcap_{j=1}^i \mathcal{H}^+(c_j, d_j)\right)$ for all $i \in [k]$.*

*Proof.* Without loss of generality, we assume that $\mathcal{R}_0 \subset \mathcal{B}_0$ where $\mathcal{B}_0$ denotes the $n$-dimensional closed $\ell_2$-ball of radius $D/2$, centered at the origin. In addition, we assume that $D > \delta$; otherwise, the statement of Claim 18 trivially follows. Let $r = \sqrt{(D^2 - \delta^2)/4}$, $\gamma = D/2 - r$, $u_x = -x/\|x\|_2$ for each $x \in bd(\mathcal{B}_0)$, and $\mathcal{S}_x^+ = \mathcal{H}^+(u_x, r)$. Then, we have

$$\mathsf{diam}(\mathcal{B}_0 \setminus \mathcal{S}_x^+) \le \delta. \tag{6}$$

Let $\mathcal{B}_1$ be an $n$-dimensional closed $\ell_2$-balls of radius $D/2 - \gamma/2 > r$, centered at the origin, i.e., $\mathcal{B}_1 \subset \mathcal{B}_0$. Since $\mathcal{B}_0 \setminus int(\mathcal{B}_1)$ is compact and $\{\mathbb{R}^n \setminus \mathcal{S}_x^+ : x \in bd(\mathcal{B}_0)\}$ is an open cover of $\mathcal{B}_0 \setminus int(\mathcal{B}_1)$, there exists a finite set $\mathcal{I} \subset bd(\mathcal{B}_0)$ such that

$$\mathcal{B}_0 \setminus int(\mathcal{B}_1) \subset \bigcup_{x \in \mathcal{I}} (\mathbb{R}^n \setminus \mathcal{S}_x^+). \tag{7}$$

Let $k = |\mathcal{I}|$, $\mathcal{I} = \{x_1, \ldots, x_k\}$, $c_i = u_{x_i}$, and $d_i = \gamma$ for all $i \in [k]$. Then by Eq. (6), it holds that

$$\mathsf{diam}(\mathcal{R}_{i-1} \setminus \mathcal{R}_i) = \mathsf{diam}(\mathcal{R}_{i-1} \setminus \mathcal{S}_{x_i}^+) \le \mathsf{diam}(\mathcal{B}_0 \setminus \mathcal{S}_{x_i}^+) \le \delta$$

for all $i \in [k]$. Furthermore, by Eq. (7) and $\mathcal{R}_0 \subset \mathcal{B}_0$, we have

$$\mathcal{R}_k = \mathcal{R}_0 \setminus \Big( \bigcup_{x \in \mathcal{I}} (\mathbb{R}^n \setminus \mathcal{S}_x^+) \Big) \subset \mathcal{B}_0 \setminus \big( \mathcal{B}_0 \setminus int(\mathcal{B}_1) \big) \subset \mathcal{B}_1.$$

This implies that

$$\mathsf{diam}(\mathcal{R}_k) \le \mathsf{diam}(\mathcal{B}_1) = D - \gamma = \frac{D}{2} + \frac{\sqrt{D^2 - \delta^2}}{2} \le D - \frac{\delta^2}{4D}$$

where the last inequality follows from the concavity of the square root: $\sqrt{a - b} \le \sqrt{a} - b/(2\sqrt{a})$ for $a \ge b > 0$. This completes the proof of Lemma 10. $\qquad \square$

### B.7 PROOF OF LEMMA 11

We first introduce the following lemmas.

**Lemma 19.** *Let $n \ge 2$, $\mathcal{P} \subset \mathbb{R}^n$ be a convex polytope, $z_1, \ldots, z_k \in \mathbb{R}^n \setminus \mathcal{P}$ be distinct points, and $\mathcal{H}^+ \subset \mathbb{R}^n$ be a closed half-space. Suppose that $z_1, \ldots, z_{l-1} \in \mathcal{H}^+$ and $z_l, \ldots, z_k \notin \mathcal{H}^+$ for some $l \in [k]$. Then, there exists a RELU network $f : \mathbb{R}^n \to \mathbb{R}^n$ of width $n$ satisfying the following:*

- *$f(x) = x$ for all $x \in \mathcal{P}$,*

- *$f(z_1), \ldots, f(z_k)$ are distinct, $f(z_i) \notin \mathcal{P}$ for all $i \in [k]$, and $f(z_1), \ldots, f(z_l) \in \mathcal{H}^+$.*

**Lemma 20.** *Let $n \ge 2$, $\mathcal{P} \subset \mathbb{R}^n$ be a convex polytope, $\mathcal{H}^+ \subset \mathbb{R}^n$ be a closed half-space such that $\mathcal{P} \cap \mathcal{H}^+ \ne \emptyset$ and $\mathcal{P} \setminus \mathcal{H}^+ \ne \emptyset$, and $v_1, \ldots, v_k \in \mathcal{H}^+ \setminus \mathcal{P}$ be distinct points. Then for any $\delta \in (0, 1)$, there exists a RELU network $f : \mathbb{R}^n \to \mathbb{R}^n$ of width $n$ satisfying the following:*

- *$f(x) = x$ for all $x \in (\mathcal{P} \cap \mathcal{H}^+) \cup \{v_1, \ldots, v_k\}$,*

- *there exist $\mathcal{S} \subset \mathcal{P} \setminus \mathcal{H}^+$ and $v_{k+1} \in \mathbb{R}^n \setminus ((\mathcal{P} \cap \mathcal{H}^+) \cup \{v_1, \ldots, v_k\})$ such that $\mu_n(\mathcal{S}) \ge \delta \cdot \mu_n(\mathcal{P} \setminus \mathcal{H}^+)$ and $f(\mathcal{S}) = \{v_{k+1}\}$.*

Consider the case $n \ge 2$. By repeatedly applying Lemma 19, one can construct a RELU network $g_1$ of width $n$ such that $g_1(x) = x$ for all $x \in \mathcal{P}$, $g_1(u_1), \ldots, g_1(u_k) \in \mathcal{H}^+ \setminus \mathcal{P}$, and $g_1(u_1), \ldots, g_1(u_k)$ are distinct. Let $v_i = g_1(u_i)$ for all $i \in [k]$. Next, we apply Lemma 20 to $\mathcal{P}$, $v_1, \ldots, v_k$, and $\mathcal{H}^+$. Then, one can construct a RELU network $g_2$ such that $g_2(x) = x$ for all $x \in (\mathcal{P} \cap \mathcal{H}^+) \cup \{v_1, \ldots, v_k\}$ and $g_2(\mathcal{S}) = \{v_{k+1}\}$ for some $v_{k+1} \notin (\mathcal{P} \cap \mathcal{H}^+) \cup \{v_1, \ldots, v_k\}$ and $\mathcal{S} \subset \mathcal{P} \setminus \mathcal{H}^+$ with $\mu_n(\mathcal{S}) \ge \delta \cdot \mu_n(\mathcal{P} \setminus \mathcal{H}^+)$. Choosing $f = g_2 \circ g_1$ completes the proof.

For the case $n = 1$, we can not directly apply Lemma 19 and Lemma 20. Nonetheless, by exploiting the inclusion map $\iota : x \mapsto (x, 0)$ for $x \in \mathbb{R}$, we can also yield the RELU network $f = g_2 \circ g_1 \circ \iota$ of width 2, which satisfies the statement of Lemma 11.

*Proof of Lemma 19.* Let $a_1 \in \mathbb{R}^n \setminus \{0\}$ and $b_1 \in \mathbb{R}$ such that $\mathcal{H}^+ = \{x \in \mathbb{R}^n : a_1^\top x + b_1 \geq 0\}$. Since $\mathcal{P}$ and $\{z_l\}$ are disjoint closed convex sets, by the hyperplane separation theorem (Boyd and Vandenberghe, 2004), there exist $a_2 \in \mathbb{R}^n \setminus \{0\}$ and $b_2 \in \mathbb{R}$ such that

$$a_2^\top x + b_2 > 0 \text{ for all } x \in \mathcal{P} \text{ and } a_2^\top z_l + b_2 < 0. \tag{8}$$

Without loss of generality, we assume that $a_2$ is not in the span of $a_1$. Otherwise, we can consider a slightly perturbed version of $a_2$ that is not in the span of $a_1$ and achieves (8); such perturbation always exists since $\mathcal{P}$ and $\{z_l\}$ are bounded.

Now, for $c \in \mathbb{R}^n \setminus \{0\}$ such that $a_2^\top c > 0$, we apply Lemma 7 to construct a ReLU network $f_c$ of width $n$ of the following form:

$$f_c(x) = \begin{cases} x & \text{if } a_2^\top x + b_2 \geq 0 \\ x - \frac{a_2^\top x + b_2}{a_2^\top c} \times c & \text{if } a_2^\top x + b_2 < 0 \end{cases}.$$

Then, from our choice of $a_2$, $b_2$, and $f_c$, (i) $f_c(x) = x$ for all $x \in \mathcal{P}$. Furthermore, consider $c \in \mathcal{S}$ where

$$\mathcal{S} \triangleq \left\{ x \in \mathbb{R}^n \setminus \{0\} : \frac{a_1^\top z_l + b_1}{a_2^\top z_l + b_2} \leq \frac{a_1^\top x}{a_2^\top x}, a_1^\top x > 0, a_2^\top x > 0 \right\}.$$

Then, we have

$$a_1^\top f_c(z_l) + b_1 = a_1^\top z_l - \frac{a_2^\top z_l + b_2}{a_2^\top c}(a_1^\top c) + b_1 \geq 0$$

where the inequality is from the definition of $\mathcal{S}$ and $a_2^\top z_l + b_2 < 0$. In addition, for any $x \in \mathbb{R}^n$, it holds that $a_1^\top f_c(x) \geq a_1^\top x$. These inequalities imply that (ii) $f_c(z_1), \ldots, f_c(z_l) \in \mathcal{H}^+$. Furthermore, we have (iii) $f_c(z_1), \ldots, f_c(z_k) \notin \mathcal{P}$: if $a_2^\top z_i + b_2 \geq 0$, then $f_c(z_i) = z_i \notin \mathcal{P} = f_c(\mathcal{P})$; otherwise, then $a_2^\top f_c(z_i) + b_2 = 0 < a_2^\top x + b_2 = a_2^\top f_c(x) + b_2$ for all $x \in \mathcal{P}$. Here, one can observe that $\mu_n(\mathcal{S}) > 0$ (i.e., $\mathcal{S}$ is non-empty) since $a_1$ and $a_2$ are linearly independent.

Lastly, we show that there exists $c \in \mathcal{S}$ such that $f_c(z_1), \ldots, f_c(z_k)$ are distinct. From the definition of $f_c$ and since $z_i \neq z_j$ for all $i \neq j$, $f_c(z_i) = f_c(z_j)$ only if $z_i - z_j$ and $c$ are linearly dependent. However, the set of vectors that is in the span of $z_i - z_j$ has zero measure with respect to $\mu_n$; however, $\mu_n(\mathcal{S}) > 0$. This implies that there exists $c \in \mathcal{S}$ such that $f_c(z_1), \ldots, f_c(z_k)$ are distinct. By (i)–(iii), choosing such $c \in \mathcal{S}$ and $f = f_c$ completes the proof. $\square$

*Proof of Lemma 20.* Without loss of generality, we assume that the normal vector of the boundary of $\mathcal{H}^+$ is $(1, 0, \ldots, 0)$, i.e., we consider

$$\mathcal{H}_1^+ = \mathcal{H}^+ = \{(x_1, \ldots, x_n) \in \mathbb{R}^n : x_1 + b_1 \geq 0\}$$

for some $b_1 \in \mathbb{R}$. Let $\mathcal{T} = (\mathcal{P} \cap \mathcal{H}_1^+) \cup \{v_1, \ldots, v_k\}$,

$$b_i = 1 - 1 \times \min_{(x_1, \ldots, x_n) \in \mathcal{T}} x_i \text{ and } \mathcal{H}_i^+ = \{(x_1, \ldots, x_n) \in \mathbb{R}^n : x_i + b_i \geq 0\}$$

for all $i \in [n] \setminus \{1\}$. We note that $b_i$ is well-defined since $\mathcal{T}$ is compact. Furthermore, from the definition of $\mathcal{H}_i^+$, it holds that $\mathcal{T} \subset \bigcap_{i=1}^n \mathcal{H}_i^+$.

For $\gamma > 0$, let $\mathcal{K}_\gamma = \mathcal{P} \cap \{(x_1, \ldots, x_n) \in \mathbb{R}^n : x_1 + b_1 + \gamma \leq 0\}$, i.e., $\mathcal{K}_\gamma \subset \mathcal{P} \setminus \mathcal{H}_1^+$. Due to the continuity of Lebesgue measure, there exists a small enough $\gamma^* > 0$ such that $\mu_n(\mathcal{K}_{\gamma^*}) \geq \delta \cdot \mu_n(\mathcal{P} \setminus \mathcal{H}_1^+)$. Here, we choose $\mathcal{S} = \mathcal{K}_{\gamma^*}$. In detail, let $\gamma_n = 1/n$ and consider corresponding $\mathcal{K}_{\gamma_n}$ and an indicator function $\mathbf{1}_{\mathcal{K}_{\gamma_n}}$. Then, our choice $\mathbf{1}_{\mathcal{K}_{\gamma_n}}$ is monotonically increasing on $\mathcal{P} \setminus \mathcal{H}_1^+$ and converges almost everywhere on $\mathcal{P} \setminus \mathcal{H}_1^+$ to $\mathbf{1}_{\mathcal{K}_0} = \mathbf{1}_{\mathcal{P} \setminus \mathcal{H}_1^+}$. Then, by Lebesgue's Monotone Convergence Theorem (Rudin, 1987), $\mu_n(K_{\gamma_n})$ converges to $\mu_n(\mathcal{P} \setminus \mathcal{H}_1^+)$. Namely, we can choose $N \in \mathbb{N}$ such that $\mu_n(\mathcal{K}_{\gamma_N}) \geq \delta \cdot \mu_n(\mathcal{P} \setminus \mathcal{H}_1^+)$.

Let $\beta_i = \sup_{(x_1, \ldots, x_n) \in \mathcal{K}_{\gamma^*}} |x_i|$ for all $i \in [n] \setminus \{1\}$; each $\beta_i$ is finite since $\mathcal{K}_{\gamma^*}$ is bounded. Now, using Lemma 7, we construct a ReLU network $f_1$ of width $n$ of the following form: for $x = (x_1, \ldots, x_n) \in \mathbb{R}^n$ and $c_1 = \left(1, -\max\{b_2 + \beta_2 + 1, 1\}/\gamma^*, \ldots, -\max\{b_n + \beta_n + 1, 1\}/\gamma^*\right)$,

$$f_1(x) = \begin{cases} x & \text{if } x_1 + b_1 \geq 0 \\ x - (x_1 + b_1)c_1 & \text{if } x_1 + b_1 < 0 \end{cases}.$$

From the definition of $f_1$, we have $f_1(x) = x \in \bigcap_{i=1}^n \mathcal{H}_i^+$ for all $x \in \mathcal{T}$. In addition, from the definition of $\mathcal{K}_{\gamma^*}$, we have $x_1 + b_1 + \gamma^* \leq 0$ for all $(x_1, \ldots, x_n) \in \mathcal{K}_{\gamma^*}$. This implies that $f_1(x)_1 = -b_1$ and

$$
\begin{aligned}
f_1(x)_i = x_i - \frac{-(x_1 + b_1)\max\{b_i + \beta_i + 1, 1\}}{\gamma^*} \\
\leq x_i - \max\{b_i + \beta_i + 1, 1\} \\
\leq \beta_i - \max\{b_i + \beta_i + 1, 1\} \\
\leq -b_i - 1
\end{aligned}
$$

for all $i \in [n] \setminus \{1\}$ and for all $x \in \mathcal{K}_{\gamma^*}$. Here, the first inequality is from $-(x_1 + b_1) \geq \gamma^* \geq 0$ and $\max\{b_i + \beta_i + 1, 1\} > 0$. We use $\beta_i \geq x_i$ for the second inequality. This implies that $f_1(\mathcal{K}_{\gamma^*}) \cap (\bigcup_{i=2}^n \mathcal{H}_i^+) = \emptyset$.

Now, we construct RELU networks $f_2, \ldots, f_n$ of width $n$ using Lemma 7 as follows: for $x = (x_1, \ldots, x_n) \in \mathbb{R}^n$ and $i \in [n] \setminus \{1\}$,

$$
f_i(x) = \begin{cases} x & \text{if } x_i + b_i \geq 0 \\ (x_1, \ldots, x_{i-1}, -b_i, x_{i+1}, \ldots, x_n) & \text{if } x_i + b_i < 0 \end{cases}.
$$

Here, one can observe that $f_i(x) = x \in \bigcap_{i=1}^n \mathcal{H}_i^+$ for all $x \in \mathcal{T}$ and $i \in [n] \setminus \{1\}$. Furthermore, since $f_1(x)_1 = -b_1$ for all $x \in \mathcal{K}_{\gamma^*}$ and $f_1(\mathcal{K}_{\gamma^*}) \cap (\bigcup_{i=2}^n \mathcal{H}_i^+) = \emptyset$, we have

$$
f_n \circ \cdots \circ f_2 \circ f_1(\mathcal{K}_{\gamma^*}) = (-b_1, -b_2, \ldots, -b_n).
$$

Since $(-b_1, \ldots, -b_n) \notin \mathcal{T}$ (e.g., $\min_{(x_1, \ldots, x_n) \in \mathcal{T}} x_2 = 1 - b_2 > -b_2$), choosing $f = f_n \circ \cdots \circ f_1$ and $v_{k+1} = (-b_1, \ldots, -b_n)$ completes the proof. $\qquad\square$

## B.8 Proof of Lemma 12

Let $\mathcal{H}_{ij} = \{x \in \mathbb{R}^n : x^\top(v_i - v_j) = 0\}$ for all $i < j$. Then, $a^\top v_1, \ldots, a^\top v_k$ are distinct if and only if $a \notin \bigcup_{i<j} \mathcal{H}_{ij}$. However, since $\mu_n(\mathcal{H}_{ij}) = 0$, we have $\mu_n(\bigcup_{i<j} \mathcal{H}_{ij}) = 0$, i.e., there exists $a \in \mathbb{R}^n \setminus (\bigcup_{i<j} \mathcal{H}_{ij})$. This completes the proof.

## C PROOF OF LOWER BOUNDS IN THEOREM 1 AND THEOREM 2

We first introduce the following lemmas.

**Lemma 21** (Lemma 1 in (Cai, 2023)). *For any activation function, networks of width $\max\{d_x, d_y\}$ are not dense in both $L^p(\mathcal{K}, \mathbb{R}^{d_y})$ and $C(\mathcal{K}, \mathbb{R}^{d_y})$.*

**Lemma 22.** *For any continuous monotone $\psi$, $\psi$ networks of width 1 are not dense in $L^p([0, 1], \mathbb{R})$.*

From Lemma 21 and Lemma 22, the proof of the lower bounds in Theorem 1 and Theorem 2 directly follow.

*Proof of Lemma 22.* In this proof, we show that for any continuous monotone $\psi$, there exists a $L^p$ measurable function $f^* : [0, 1] \to \mathbb{R}$ that cannot be approximated by any $\psi$ network of width 1, say $f$, within $1/6$ error measured by $L^p$ norm. Here, by the Hölder's inequality, it suffices to show that $\|f - f^*\|_1 \geq 1/6$.

Since any compositions of continuous monotone functions are continuous monotone, without loss of generality, we assume that a $\varphi$ network $f$ is a continuous and monotonically increasing function.

Consider $f^* : [0, 1] \to \mathbb{R}$ defined as

$$f^*(x) = \begin{cases} 0 & \text{if } x \in [0, 1/3] \cup [2/3, 1] \\ 1 & \text{if } x \in (1/3, 2/3) \end{cases},$$

and let $f(2/3) = c$ for some $c \in \mathbb{R}$. Then if $c \leq 0$,

$$\int_{[0,1]} |f - f^*| dx \geq \int_{[1/3,2/3]} |f - f^*| dx \geq \int_{[1/3,2/3]} |f^*| dx = \frac{1}{3}.$$

Likewise, if $c \geq 1$,

$$\int_{[0,1]} |f - f^*| dx \geq \int_{[2/3,1]} |f - f^*| dx \geq \int_{[2/3,1]} |1 - f^*| = \frac{1}{3}.$$

Furthermore, if $c \in (0, 1)$,

$$\int_{[0,1]} |f - f^*| dx \geq \int_{[1/3,2/3]} |f - f^*| dx + \int_{[2/3,1]} |f - f^*| dx$$

$$\geq \int_{[1/3,2/3]} |c - f^*| dx + \int_{[2/3,1]} |c - f^*| dx = \frac{1}{3}(1 - c) + \frac{1}{3}c = \frac{1}{3}.$$

Hence, for any continuous monotone $\varphi$ network can not approximate $f^*$ within $1/6$ error, which completes the proof. □

# D    PROOF OF UPPER BOUND IN THEOREM 2

## D.1    ADDITIONAL NOTATIONS

Throughout this section, we use RELU-LIKE for the set of RELU-like activation functions of our interest, defined as

$$\text{RELU-LIKE} \triangleq \{\text{SOFTPLUS}, \text{Leaky-RELU}, \text{ELU}, \text{CELU}, \text{SELU}, \text{GELU}, \text{SILU}, \text{MISH}\}.$$

For $n \in \mathbb{N}$ and $f : \mathbb{R} \to \mathbb{R}$, we denote $f^n(x) \triangleq f \circ \cdots \circ f$ the $n$-th iterate of the function $f$. For any continuous function $f$ on a compact domain $\mathcal{K} \subset \mathbb{R}^n$, $\omega_{p,f}$ denotes the modulus of continuity of $f$ in the $p$-norm: $\|f(x) - f(x')\|_p \leq \omega_{p,f}(\|x - x'\|_p)$ for all $x, x' \in \mathcal{K}$. We note that such $\omega_{p,f}$ is well-defined on a compact domain since $f$ is uniformly continuous on a compact domain.

## D.2    PROOF OF UPPER BOUND IN THEOREM 2

In this proof, we show that for any $\varepsilon > 0$, $\varphi \in$ RELU-LIKE, and RELU network $f$ of width $w$, there exists a $\varphi$ network $g$ with the same width $w$ such that

$$\|f - g\|_p \leq \varepsilon.$$

Then, combining the above bound and the upper bound $w_{\min} \leq \{d_x, d_y, 2\}$ in Theorem 1 completes the proof of the upper bound in Theorem 2.

To this end, we first introduce the following lemma. The proof of Lemma 23 is presented in Appendix D.3.

**Lemma 23.** *For any given compact set $\mathcal{K} \subset \mathbb{R}$ and activation function $\varphi \in$ RELU-LIKE, there exists a sequence $\{h_n\}$ of $\varphi$ networks of width $1$ such that it uniformly converges to RELU on $\mathcal{K}$.*

Lemma 23 states that for any given compact $\mathcal{K} \subset \mathbb{R}$, for each RELU-like activation function $\varphi$, and some fixed $\delta > 0$, there exist a sequence $\{h_n\}$ of $\varphi$ networks of width $1$ and $N \in \mathbb{N}$ such that $\|\text{RELU}(x) - h_N(x)\|_\infty \leq \delta$ on all $n \geq N$ and $x \in \mathcal{K}$; we will assign an explicit value to $\delta$ later. We note that it suffices to show uniform convergence on an arbitrary compact set $\mathcal{K}$ since functions of our interests are defined on compact domains.

Here, we denote a RELU network $f$ as below, recalling (1):

$$f = t_L \circ \phi_{L-1} \circ \cdots \circ t_2 \circ \phi_1 \circ t_1$$

where $L \in \mathbb{N}$ is the number of layers, $t_\ell : \mathbb{R}^{d_{\ell-1}} \to \mathbb{R}^{d_\ell}$ is an affine transformation, and $\phi_\ell(x_1, \ldots, x_{d_\ell}) = (\text{RELU}(x_1), \ldots, \text{RELU}(x_{d_\ell}))$ for all $\ell \in [L]$. And, for each RELU-like activation function $\varphi$, we choose a $\varphi$ network $g$ via applying the same affine maps $t_1, \ldots, t_L$ and $h_n$, which is satisfying Lemma 23, such that

$$g = t_L \circ \rho_{L-1} \circ \cdots \circ t_2 \circ \rho_1 \circ t_1$$

where $\rho_\ell(x_1, \ldots, x_{d_\ell}) = (h_n(x_1), \ldots, h_n(x_{d_\ell}))$ for all $\ell \in [L]$. We further denote $f_\ell$ and $g_\ell$ by the first $\ell - 1$ layers of $f$ and $g$ with the subsequent affine layer $t_\ell$, respectively:

$$f_\ell = t_\ell \circ \phi_{\ell-1} \circ \cdots \circ \phi_1 \circ t_1 \quad \text{and} \quad g_\ell = t_\ell \circ \rho_{\ell-1} \circ \cdots \circ \rho_1 \circ t_1$$

Then, for each $\ell \in [L] \setminus \{1\}$, we have

$$
\begin{aligned}
\|f_\ell - g_\ell\|_p &= \|t_\ell \circ \phi_{\ell-1} \circ f_{\ell-1} - t_\ell \circ \rho_{\ell-1} \circ g_{\ell-1}\|_p \\
&\leq \omega_{t_\ell,p}\left(\|\phi_{\ell-1} \circ f_{\ell-1} - \rho_{\ell-1} \circ g_{\ell-1}\|_p\right) \\
&\leq \omega_{t_\ell,p}\left(\|\phi_{\ell-1} \circ f_{\ell-1} - \phi_{\ell-1} \circ g_{\ell-1}\|_p + \|\phi_{\ell-1} \circ g_{\ell-1} - \rho_{\ell-1} \circ g_{\ell-1}\|_p\right) \\
&\leq \omega_{t_\ell,p}\Bigg(\|\phi_{\ell-1} \circ f_{\ell-1} - \phi_{\ell-1} \circ g_{\ell-1}\|_p \\
&\qquad\qquad + \left(\int_{[0,1]^{d_x}} \|\phi_{\ell-1} \circ g_{\ell-1}(x) - \rho_{\ell-1} \circ g_{\ell-1}(x)\|_p^p d\mu_{d_x}\right)^{1/p}\Bigg) \\
&= \omega_{t_\ell,p}\Bigg(\|\phi_{\ell-1} \circ f_{\ell-1} - \phi_{\ell-1} \circ g_{\ell-1}\|_p \\
&\qquad\qquad + \left(\int_{[0,1]^{d_x}} \sum_{i=1}^{d_{\ell-1}} \left(\text{RELU}(g_{\ell-1}(x)_i) - h_n(g_{\ell-1}(x)_i)\right)^p d\mu_{d_x}\right)^{1/p}\Bigg)
\end{aligned}
$$

We note that $\omega_{t_\ell,p}$ is well-defined on $[0,1]^{d_x}$ since $t_\ell$ is uniformly continuous on $[0,1]^{d_x}$.

For each $i \in [d_{\ell-1}]$, by Lemma 23, there exists $N_i \in \mathbb{N}$ such that

$$
\|\text{RELU}(g_{\ell-1}(x)_i) - h_n(g_{\ell-1}(x)_i)\|_\infty \leq \delta
$$

for all $n \geq N_i$ and $x \in [0,1]^{d_x}$. Moreover, from the definition of $\phi_{\ell-1}$, we have

$$
\|\phi_{\ell-1} \circ f_{\ell-1} - \phi_{\ell-1} \circ g_{\ell-1}\|_p \leq \omega_{\phi_{\ell-1},p}(\|f_{\ell-1} - g_{\ell-1}\|_p) \leq \|f_{\ell-1} - g_{\ell-1}\|_p.
$$

Therefore, for $n \geq \max\{N_1, \ldots, N_{d_{\ell-1}}\}$, we have

$$
\begin{aligned}
\|f_\ell - g_\ell\|_p &\leq \omega_{t_\ell,p}\Bigg(\|\phi_{\ell-1} \circ f_{\ell-1} - \phi_{\ell-1} \circ g_{\ell-1}\|_p \\
&\qquad\qquad + \left(\int_{[0,1]^{d_x}} \sum_{i=1}^{d_{\ell-1}} \left(\text{RELU}(g_{\ell-1}(x)_i) - h_n(g_{\ell-1}(x)_i)\right)^p d\mu_{d_x}\right)^{1/p}\Bigg) \\
&\leq \omega_{t_\ell,p}\left(\|f_{\ell-1} - g_{\ell-1}\|_p + \delta \times d_{\ell-1}^{1/p}\right), \tag{9}
\end{aligned}
$$

with $\|f_1 - g_1\|_p = \|t_1 - t_1\|_p = 0$.

Consequently, by iteratively applying (9), we get

$$
\begin{aligned}
\|f - g\|_p &\leq \omega_{t_L,p}\left(\|f_{L-1} - g_{L-1}\|_p + \delta \times d_{L-1}^{1/p}\right) \\
&\leq \omega_{t_L,p}\left(\omega_{t_{L-1},p}\left(\|f_{L-2} - g_{L-2}\|_p + \delta \times d_{L-2}^{1/p}\right) + \delta \times d_{L-1}^{1/p}\right) \\
&\;\;\vdots \\
&\leq \omega_{t_L,p}\left(\omega_{t_{L-1},p}\left(\cdots \omega_{t_2,p}\left(\|f_1 - g_1\|_p + \delta \times d_1^{1/p}\right)\cdots + \delta \times d_{L-2}^{1/p}\right) + \delta \times d_{L-1}^{1/p}\right) \\
&= \omega_{t_L,p}\left(\omega_{t_{L-1},p}\left(\cdots \omega_{t_2,p}\left(\delta \times d_1^{1/p}\right)\cdots + \delta \times d_{L-2}^{1/p}\right) + \delta \times d_{L-1}^{1/p}\right). \tag{10}
\end{aligned}
$$

Thus, we can bound the right-hand side (10) of the above inequality within any $\varepsilon > 0$, by choosing sufficiently small $\delta > 0$. Hence, it completes the proof of the upper bound in Theorem 2.

### D.3 PROOF OF LEMMA 23

In this section, we explicitly construct a sequence of $\varphi$ network $\{h_n\}$ satisfying Lemma 23. Namely, we show that for any $\varepsilon > 0$, for any compact set $\mathcal{K} \subset \mathbb{R}$, there exists $N \in \mathbb{N}$ such that $|h_n(x) - \text{RELU}(x)| \leq \varepsilon$ for all $n \geq N$ and $x \in \mathcal{K}$. Without loss of generality, we assume that $\mathcal{K} = [-m, M]$ for some $m, M > 0$.

1. $\varphi = $ SOFTPLUS:

In this case, we claim that $h_n(x) = (t_2 \circ \varphi \circ t_1)(x)$ completes the proof for SOFTPLUS, where $t_1(x) = nx$ and $t_2(x) = x/n$. If $x \geq 0$, by the Mean Value Theorem, we have

$$|h_n(x) - \text{RELU}(x)| = \frac{1}{\beta n} \log(1 + \exp(\beta nx)) - \frac{1}{\beta n} \log(\exp(\beta nx))$$
$$\leq \frac{1}{\beta n}.$$

Otherwise, if $x < 0$,

$$|h_n(x) - \text{RELU}(x)| = \frac{1}{\beta n} \log(1 + \exp(\beta nx)) < \frac{1}{\beta n} \log(2)$$

since SOFTPLUS is strictly increasing. Hence, choosing sufficiently large $N \in \mathbb{N}$ such that $1/\beta N \leq \varepsilon$, which completes the proof for SOFTPLUS.

2. $\varphi = $ Leaky-RELU:

In this case, we claim that $h_n(x) = \varphi^n(x)$ completes the proof for Leaky-RELU. Here, from the definition of Leaky-RELU, we only consider for $x < 0$. Then,

$$|h_n(x) - \text{RELU}(x)| = \alpha^n |x| \leq \alpha^n \times m$$

We note that $\alpha \in (0, 1)$. Hence, choosing sufficiently large $N \in \mathbb{N}$ such that $\alpha^N \times m \leq \varepsilon$, which completes the proof for Leaky-RELU.

3. $\varphi = $ ELU:

Similar to the case $\varphi = $ leaky-RELU, we claim that $h_n(x) = \varphi^n(x)$ completes the proof for ELU and only consider for $x < 0$ from the definition of ELU. Since ELU is bounded below by $-\alpha$, strictly increasing, and $\text{ELU}(0) = 0$, we have

$$|h_1(x)| < \alpha \Rightarrow |h_2(x)| < \alpha(1 - \exp(-\alpha)) < \alpha$$
$$\Rightarrow |h_3(x)| < \alpha(1 - \exp(-\alpha(1 - \exp(-\alpha)))) < \alpha(1 - \exp(-\alpha))$$

We note that the upper bound of the sequence $\{|h_n|\}$ is strictly decreasing and its infimum is equal to 0. Thus, by the monotone convergence theorem, there exists $N \in \mathbb{N}$ such that $|h_n(x) - \text{RELU}(x)| = |h_n(x)| \leq \varepsilon$ for all $n \geq N$. Hence, this completes the proof for ELU.

4. $\varphi = $ CELU:

Since CELU is a smooth variant of ELU, the proof technique for CELU is the same as ELU. Consider $h_n(x) = \varphi^n(x)$. Then, for $x < 0$, we have

$$|h_1(x)| < \alpha \Rightarrow |h_2(x)| < \alpha(1 - 1/e) < \alpha$$
$$\Rightarrow |h_3(x)| < \alpha(1 - \exp(1/e - 1)) < \alpha(1 - 1/e)$$

Likewise, we note that the upper bound of the sequence $\{|h_n|\}$ is strictly decreasing and its infimum is equal to 0. Thus, by the monotone convergence theorem, there exists $N \in \mathbb{N}$ such that $|h_n(x) - \text{RELU}(x)| = |h_n(x)| \leq \varepsilon$ for all $n \geq N$. Hence, this completes the proof for CELU.

5. $\varphi = $ SELU:

From the definition of ELU and SELU, we can represent SELU as $\lambda \times \text{ELU}$. Hence, from the proof for ELU, $h_n(x) = (t_\lambda \circ \varphi)^n(x)$ completes the proof for SELU, where $t_\lambda(x) = x/\lambda$.

6. $\varphi = $ GELU:

In this case, we claim that $h_n(x) = (t_2 \circ \varphi \circ t_1)(x)$ completes the proof for GELU, where $t_1(x) = nx$ and $t_2(x) = x/n$. If $x \geq 0$, we have

$$|\varphi_n(x) - \text{RELU}(x)| = x(1 - \Phi(nx)) \leq x \exp(-n^2 x^2/2) \leq \frac{1}{n\sqrt{e}}$$

since $P(Z \geq t) \leq \exp(-t^2/2)$ for all $t \geq 0$ where $Z \sim \mathcal{N}(0, 1)$. We note that the last inequality is derived from its derivative. Otherwise, if $x < 0$,

$$|\varphi_n(x) - \text{RELU}(x)| = x\Phi(nx) \leq x \exp(-n^2 x^2/2) \leq \frac{1}{n\sqrt{e}}$$

since $P(Z \leq t) \leq \exp(-t^2/2)$ for all $t \leq 0$ where $Z \sim \mathcal{N}(0,1)$. Hence, choosing sufficiently large $N \in \mathbb{N}$ such that $1/N \leq \varepsilon$, which completes the proof for GELU.

7. $\varphi = \text{SILU}$:

In this case, we claim that $h_n(x) = (t_2 \circ \varphi \circ t_1)(x)$ completes the proof for SILU, where $t_1(x) = nx$ and $t_2(x) = x/n$. If $x \geq 0$, we have

$$|\varphi_n(x) - \text{RELU}(x)| = x \left( 1 - \frac{1}{1 + \exp(-nx)} \right) = \frac{x \exp(-nx)}{1 + \exp(-nx)}$$
$$< x \exp(-nx) \leq \frac{1}{en}.$$

Note that the last inequality is derived from its derivative. Otherwise, if $x < 0$,

$$|\varphi_n(x) - \text{RELU}(x)| = \frac{-x}{1 + \exp(-nx)} \leq \frac{1}{n}.$$

since $1 - x \leq \exp(-x)$ for all $x \in \mathbb{R}$. Hence, choosing sufficiently large $N \in \mathbb{N}$ such that $1/N \leq \varepsilon$, which completes the proof for SILU.

8. $\varphi = \text{MISH}$:

In this case, we claim that $h_n(x) = (t_2 \circ \varphi \circ t_1)(x)$ completes the proof for MISH, where $t_1(x) = nx$ and $t_2(x) = x/n$. If $x \geq 0$, we have

$$|\varphi_n(x) - \text{RELU}(x)| = x \left( 1 - \frac{(1 + \exp(nx))^2 - 1}{(1 + \exp(nx))^2 + 1} \right)$$
$$= \frac{2x}{(1 + \exp(nx))^2 + 1} \leq \frac{x}{1 + \exp(nx)} \leq \frac{1}{n}.$$

since $1 + x \leq \exp(x)$ for all $x \in \mathbb{R}$. Otherwise, if $x < 0$,

$$|\varphi_n(x) - \text{RELU}(x)| = -x \times \frac{(1 + \exp(nx))^2 - 1}{(1 + \exp(nx))^2 + 1}$$
$$\leq -x \times \frac{(1 + \exp(nx))^2 - 1}{2 + \exp(nx)}$$
$$= -x \exp(nx) \leq \frac{1}{en}.$$

Note that the last inequality is derived from its derivative. Hence, choosing sufficiently large $N \in \mathbb{N}$ such that $1/N \leq \varepsilon$, which completes the proof for MISH.

# E  PROOF OF LEMMA 8

In this proof, we show that for any $\varepsilon > 0$, $\sigma$ that can be uniformly approximated by some sequence of continuous injection, say $\varphi_n$, and $\sigma$ network $f$ of width $w$, there exists $\varphi_n$ network $g$ of the same width $w$ such that

$$\|f - g\|_\infty \le \varepsilon.$$

From the assumption, there exists a sequence of continuous injection $\{\varphi_n\}$ that uniformly converges to $\sigma$ on $\mathbb{R}$. Namely, for any $\delta > 0$, there exists $N \in \mathbb{N}$ such that $\|\varphi_n(x) - \sigma(x)\|_\infty \le \delta$ for all $n \ge N$ and $x \in \mathbb{R}$; we will assign an explicit value to $\delta$ later.

Now, we denote a $\sigma$ network $f$ as below, recalling (1):

$$f = t_L \circ \phi_{L-1} \circ \cdots \circ t_2 \circ \phi_1 \circ t_1$$

where $L \in \mathbb{N}$ is the number of layers, $t_\ell : \mathbb{R}^{d_{\ell-1}} \to \mathbb{R}^{d_\ell}$ is an affine transformation, and $\phi_\ell(x_1, \ldots, x_{d_\ell}) = (\sigma(x_1), \ldots, \sigma(x_{d_\ell}))$ for all $\ell \in [L]$. And, we choose a $\varphi_n$ network such that

$$g = t_L \circ \rho_{L-1} \circ \cdots \circ t_2 \circ \rho_1 \circ t_1$$

where $\rho_\ell(x_1, \ldots, x_{d_\ell}) = (\varphi_n(x_1), \ldots, \varphi_n(x_{d_\ell}))$ for all $\ell \in [L]$. We further denote $f_\ell$ and $g_\ell$ by the first $\ell - 1$ layers of $f$ and $g$ with the subsequent affine layer $t_\ell$, respectively:

$$f_\ell = t_\ell \circ \phi_{\ell-1} \circ \cdots \circ \phi_1 \circ t_1 \quad \text{and} \quad g_\ell = t_\ell \circ \rho_{\ell-1} \circ \cdots \circ \rho_1 \circ t_1$$

Then, for each $\ell \in [L] \setminus \{1\}$, we have

$$\|f_\ell - g_\ell\|_\infty = \|t_\ell \circ \phi_{\ell-1} \circ f_{\ell-1} - t_\ell \circ \rho_{\ell-1} \circ g_{\ell-1}\|_\infty$$

$$\le \omega_{t_\ell,\infty} \left( \|\phi_{\ell-1} \circ f_{\ell-1} - \rho_{\ell-1} \circ g_{\ell-1}\|_\infty \right)$$

$$\le \omega_{t_\ell,\infty} \left( \|\phi_{\ell-1} \circ f_{\ell-1} - \phi_{\ell-1} \circ g_{\ell-1}\|_\infty + \|\phi_{\ell-1} \circ g_{\ell-1} - \rho_{\ell-1} \circ g_{\ell-1}\|_\infty \right)$$

$$\le \omega_{t_\ell,\infty} \Big( \|\phi_{\ell-1} \circ f_{\ell-1} - \phi_{\ell-1} \circ g_{\ell-1}\|_\infty$$

$$+ \sup_{x \in [0,1]^{d_x}} \|\phi_{\ell-1} \circ f_{\ell-1}(x) - \rho_{\ell-1} \circ g_{\ell-1}(x)\|_\infty \Big)$$

$$= \omega_{t_\ell,\infty} \Big( \|\phi_{\ell-1} \circ f_{\ell-1} - \phi_{\ell-1} \circ g_{\ell-1}\|_\infty$$

$$+ \sup_{x \in [0,1]^{d_x}} \max_{i \in [d_{\ell-1}]} \{\sigma(g_{\ell-1}(x)_i) - \varphi_n(g_{\ell-1}(x)_i)\} \Big).$$

We note that $\omega_{t_\ell,\infty}$ is well-defined on $[0,1]^{d_x}$ since $t_\ell$ is uniformly continuous on $[0,1]^{d_x}$.

Since $\sigma$ is uniformly approximated by $\varphi_n$, for each $i \in [d_{\ell-1}]$, there exists $N_i \in \mathbb{N}$ such that

$$\|\sigma(g_{\ell-1}(x)_i) - \varphi_n(g_{\ell-1}(x)_i)\|_\infty \le \delta$$

for all $n \ge N_i$ and $x \in [0,1]^{d_x}$. Therefore, for $n \ge \max\{N_1, \ldots, N_{d_{\ell-1}}\}$, we have

$$\|f_\ell - g_\ell\|_\infty \le \omega_{t_\ell,\infty} \Big( \|\phi_{\ell-1} \circ f_{\ell-1} - \phi_{\ell-1} \circ g_{\ell-1}\|_\infty$$

$$+ \sup_{x \in [0,1]^{d_x}} \max_{i \in [d_{\ell-1}]} \{\sigma(g_{\ell-1}(x)_i) - \varphi_n(g_{\ell-1}(x)_i)\} \Big)$$

$$\le \omega_{t_\ell,\infty} \left( \omega_{\phi_{\ell-1},\infty}(\|f_{\ell-1} - g_{\ell-1}\|_\infty) + \delta \right), \tag{11}$$

with $\|f_1 - g_1\|_p = \|t_1 - t_1\|_p = 0$. Again, note that $\omega_{\phi_{\ell-1},\infty}$ is well-defined on $[0,1]^{d_x}$ since $\varphi_{\ell-1}$ is uniformly continuous on $[0,1]^{d_x}$.

Consequently, by iteratively applying (11), we get

$$\|f - g\|_\infty \le \omega_{t_L,\infty} \left( \omega_{\phi_{L-1},\infty}(\|f_{L-1} - g_{L-1}\|_\infty) + \delta \right)$$

$$\le \omega_{t_L,\infty} \left( \omega_{\phi_{L-1},\infty} \left( \omega_{t_{L-1},\infty} \left( \omega_{\phi_{L-2},\infty}(\|f_{L-2} - g_{L-2}\|_\infty) + \delta \right) \right) + \delta \right)$$

$$\vdots$$

$$\le \omega_{t_L,\infty} \left( \omega_{\phi_{L-1},\infty} \left( \cdots \left( \omega_{t_3,\infty} \left( \omega_{\phi_2,\infty}(\|f_2 - g_2\|_\infty) + \delta \right) \right) \cdots \right) + \delta \right)$$

$$\le \omega_{t_L,\infty} \left( \omega_{\phi_{L-1},\infty} \left( \cdots \left( \omega_{t_3,\infty} \left( \omega_{\phi_2,\infty} \left( \omega_{t_2,\infty}(\delta) \right) + \delta \right) \right) \cdots \right) + \delta \right) \tag{12}$$

Therefore, we can bound the right-hand side (12) of the above inequality within arbitrary $\varepsilon > 0$, by choosing sufficiently small $\delta > 0$. Hence, it completes the proof of Lemma 8.

## F    Minimum width for $L^p$ approximation of RNNs

In Theorems 1 and 2, we prove the exact minimum width for $L^p$ approximation via networks using ReLU or ReLU-Like activation functions. Using similar proof techniques, we investigate the minimum width for $L^p$ approximation for other network architectures: recurrent neural networks (RNNs) and bidirectional RNNs (BRNNs) in this section.

### F.1    Additional notations

We first introduce additional notations that will be used throughout this section. Given a length $T$ sequence of $d$-dimensional vectors $x \in \mathbb{R}^{d \times T}$ (i.e., $x$ is a matrix of $d$ rows and $T$ columns), we denote a token at index $t \in [T]$ by $x[t] \in \mathbb{R}^d$ (i.e., the $t$-th column of $x$) and tokens from index $t_1 \in [T]$ to $t_2 \in [T]$ by $x[t_1 : t_2] \in \mathbb{R}^{d \times (t_2 - t_1 + 1)}$ for $t_1 < t_2$ (i.e., the submatrix of $x$ consisting of its $t_1$-th,...,$t_2$-th columns). We define recurrent cells used in RNN and BRNN architectures as follows.

- **RNN cell.** A recurrent cell $\vec{R}_\ell$ of the layer $\ell$ with hidden dimension $d_\ell$ maps an input sequence $x = (x[1], \ldots, x[T]) \in \mathbb{R}^{d_\ell \times T}$ to an output sequence $y = (y[1], \ldots, y[T]) \in \mathbb{R}^{d_\ell \times T}$ such that

$$y[t + 1] = \vec{R}_\ell(x)[t + 1] \triangleq \phi_\ell(\vec{W}_{\ell,1} \vec{R}_\ell(x)[t] + \vec{W}_{\ell,2} x[t + 1] + \vec{b}_\ell),$$

  where $\phi_\ell(x_1, \ldots, x_{d_\ell}) = (\sigma(x_1), \ldots, \sigma(x_{d_\ell}))$ is a coordinate-wise activation function, and $\vec{W}_{\ell,1}, \vec{W}_{\ell,2} \in \mathbb{R}^{d_\ell \times d_\ell}$ and $\vec{b}_\ell \in \mathbb{R}^{d_\ell}$ are the weight parameters. The initial hidden state $\vec{R}_\ell(x)[0]$ is set to be $0 \in \mathbb{R}^{d_\ell}$.

- **BRNN cell.** A bidirectional recurrent cell $\overleftrightarrow{R}_\ell$ of the layer $\ell$ with hidden dimension $d_\ell$ consists of a pair of recurrent cells $\vec{R}_\ell, \overleftarrow{R}_\ell$ with the same hidden dimension, and additional weight parameters $A_\ell, B_\ell \in \mathbb{R}^{d_\ell \times d_\ell}$ such that

$$\vec{R}_\ell(x)[t + 1] = \phi_\ell(\vec{W}_{\ell,1} \vec{R}_\ell(x)[t] + \vec{W}_{\ell,2} x[t + 1] + \vec{b}_\ell),$$
$$\overleftarrow{R}_\ell(x)[t - 1] \triangleq \phi_\ell(\overleftarrow{W}_{\ell,1} \overleftarrow{R}_\ell(x)[t] + \overleftarrow{W}_{\ell,2} x[t - 1] + \overleftarrow{b}_\ell),$$
$$y[t + 1] = \overleftrightarrow{R}_\ell(x)[t] \triangleq A_\ell \vec{R}_\ell(x)[t] + B_\ell \overleftarrow{R}_\ell(x)[t],$$

  where the initial hidden states $\vec{R}_\ell(x)[0]$ and $\overleftarrow{R}_\ell(x)[T + 1]$ are set to be $0 \in \mathbb{R}^{d_\ell}$.

- **Network architecture.** Given an activation function $\sigma : \mathbb{R} \to \mathbb{R}$, token-wise linear maps $P : \mathbb{R}^{d_x \times T} \to \mathbb{R}^{d \times T}$ and $Q : \mathbb{R}^{d \times T} \to \mathbb{R}^{d_y \times T}$ (i.e., there are some linear maps $\phi : \mathbb{R}^{d_x} \to \mathbb{R}^d$ and $\psi : \mathbb{R}^d \to \mathbb{R}^{d_y}$ such that $P(x)[t] = \phi(x[t])$ and $Q(x)[t] = \psi(x[t])$), and $L$ recurrent cells $\vec{R}_1, \ldots, \vec{R}_L$ with hidden dimensions $d_1, \ldots, d_L$, we define an RNN $f$ as follows:

$$f \triangleq Q \circ \vec{R}_L \circ \cdots \circ \vec{R}_1 \circ P.$$

We denote a neural network $f$ with an activation function $\sigma$ by a "$\sigma$ RNN". If we replace RNN cells $\vec{R}_1, \ldots, \vec{R}_L$ to BRNN cells $\overleftrightarrow{R}_1, \ldots, \overleftrightarrow{R}_L$, then we denote a function $f$ by a "$\sigma$ BRNN". We define the width of RNN (or BRNN) $f$ as the maximum over $d_1, \ldots, d_L$.

We now introduce function spaces to universally approximate via RNNs and BRNNs. Given $T \in \mathbb{N}$, we define the target function class $L^p(\mathcal{X}^T, \mathcal{Y}^T)$, which consists of all $L^p$ sequence-to-sequence functions with length $T$ from $\mathcal{X} \subset \mathbb{R}^{d_x}$ to $\mathcal{Y} \subset \mathbb{R}^{d_y}$, endowed with the entry-wise $L^p$-norm: $\|f\|_{p,p} \triangleq (\int_{\mathcal{X}^T} \|f(x)\|_{p,p}^p dx)^{1/p}$ where $\|\cdot\|_{p,p}$ is the $L_{p,p}$ norm, i.e., an entry-wise matrix norm. Unlike BRNNs, output tokens of RNNs at index $t \in [T]$ only depend on $x[1 : t] \in \mathbb{R}^{d_x \times t}$. We refer to such functions that only depend on past information as *the past-dependent functions*. Namely, a function $f : \mathbb{R}^{d_1 \times T} \to \mathbb{R}^{d_2 \times T}$ is past-dependent if

$$f(x)[t] = g_t(x[1 : t])$$

for some $g_t : \mathbb{R}^{d_1 \times t} \to \mathbb{R}^{d_2}$ for all $t \in [T]$. For a target function class for universal approximation using RNNs, we consider past-dependent $L^p([0, 1]^{d_x \times T}, \mathbb{R}^{d_y \times T})$, which is a space of all past-dependent functions $f$ such that $f \in L^p([0, 1]^{d_x \times T}, \mathbb{R}^{d_y \times T})$. For a target function class for universal approximation using BRNNs, we consider $L^p([0, 1]^{d_x \times T}, \mathbb{R}^{d_y \times T})$.

Table 2: A known bounds on the minimum width for $L^p$ approximation via RNNs and BRNNs using RELU or RELU-LIKE activation functions. In this table, $p \in [1, \infty)$ and all results with the domain $[0,1]^{d_x \times T}$ extends to $\mathcal{K}^T$ where $\mathcal{K}$ denotes an arbitrary compact set in $\mathbb{R}^{d_x}$.

| Reference | Network | Function class | Activation $\sigma$ | Upper/lower bounds |
|---|---|---|---|---|
| Song et al. (2023) | RNN | $L^p(\mathbb{R}^{d_x \times T}, \mathbb{R}^{d_y \times T})^{\P}$ | RELU | $w_{\min} = \max\{d_x + 1, d_y\}$ |
| | BRNN | $L^p(\mathbb{R}^{d_x \times T}, \mathbb{R}^{d_y \times T})$ | RELU | $w_{\min} \leq \max\{d_x + 1, d_y\}$ |
| Theorem 24 Theorem 25 | RNN | $L^p([0,1]^{d_x \times T}, \mathbb{R}^{d_y \times T})^{\P}$ | RELU | $w_{\min} = \max\{d_x, d_y, 2\}$ |
| | | | RELU-LIKE$^{\S}$ | $w_{\min} = \max\{d_x, d_y, 2\}$ |
| Theorem 26 | BRNN | $L^p([0,1]^{d_x \times T}, \mathbb{R}^{d_y \times T})$ | RELU | $w_{\min} \leq \max\{d_x, d_y, 2\}$ |
| | | | RELU-LIKE$^{\|}$ | $w_{\min} \leq \max\{d_x, d_y, 2\}$ |

¶ requires the class to consist of past-dependent functions.
§ includes SOFTPLUS, Leaky-RELU, ELU, CELU, SELU, GELU, SILU, and MISH where GELU, SILU, and MISH requires $d_x + d_y \geq 3$.
‖ do not require $d_x + d_y \geq 3$ for GELU, SILU, and MISH.

Before describing our results, we introduce a recent work for universal approximation of RNNs (Song et al., 2023). Song et al. (2023) show that the upper bound on the minimum width for universal approximation is independent of the length of the input sequences. In particular, they consider unbounded domain and prove that width $\max\{d_x + 1, d_y\}$ is necessary and sufficient for RELU RNNs to be dense in the past-dependent $L^p(\mathbb{R}^{d_x \times T}, \mathbb{R}^{d_y \times T})$ and the same width $\max\{d_x + 1, d_y\}$ is sufficient for RELU BRNNs to universally approximate $L^p(\mathbb{R}^{d_x \times T}, \mathbb{R}^{d_y \times T})$.

## F.2 OUR RESULTS

We are now ready to introduce our results on a compact domain. The first result characterizes the exact minimum width of RNNs to be dense in past-dependent $L^p([0,1]^{d_x \times T}, \mathbb{R}^{d_y \times T})$. The proof of Theorems 24 and 25 are presented in Appendix F.3 and Appendix F.4, respectively.

**Theorem 24.** *For any $T \in \mathbb{N}$, $w_{\min} = \{d_x, d_y, 2\}$ for RELU RNNs to be dense in past-dependent $L^p([0,1]^{d_x \times T}, \mathbb{R}^{d_y \times T})$.*

**Theorem 25.** *For any $T \in \mathbb{N}$, $w_{\min} = \{d_x, d_y, 2\}$ for $\varphi$ RNNs to be dense in past-dependent $L^p([0,1]^{d_x \times T}, \mathbb{R}^{d_y \times T})$ if $\varphi \in \{\text{ELU}, \text{Leaky-RELU}, \text{SOFTPLUS}, \text{CELU}, \text{SELU}\}$, or $\varphi \in \{\text{GELU}, \text{SILU}, \text{MISH}\}$ and $d_x + d_y \geq 3$.*

Theorems 24 and 25 characterize the minimum width of RNNs using RELU or RELU-LIKE activation functions to be dense in past-dependent $L^p([0,1]^{d_x \times T}, \mathbb{R}^{d_y \times T})$ is exactly $\max\{d_x, d_y, 2\}$, which coincides with the fully-connected network case (Theorems 1 and 2). Further, Theorem 24 shows a dichotomy between the minimum width of RELU RNNs for $L^p$ approximation on the compact domain and the whole Euclidean space. A similar observation also holds for RNNs using RELU-LIKE activation functions using Theorem 25.

In order to prove the upper bound $w_{\min} \leq \{d_x, d_y, 2\}$ in Theorems 24 and 25, we use coding-based proof techniques as in (Song et al., 2023) but with different coding schemes (e.g., as in Lemma 5). The lower bound $w_{\min} \geq \{d_x, d_y, 2\}$ in Theorem 24 directly follows from the facts that for any $\varphi \in \{\text{RELU}\} \cup \text{RELU-LIKE}$ and $\varphi$ RNN $f$, $f(x)[1] = Q \circ \vec{R}_L \circ \cdots \circ \vec{R}_1 \circ P(x)[1]$ is a $\varphi$ network and $w_{\min} \geq \max\{d_x, d_y, 2\}$ is necessary for $\varphi$ networks to be dense in $L^p([0,1]^{d_x}, \mathbb{R}^{d_y})$ (Theorems 1 and 2).

Our next result shows that the same upper bound in Theorem 24 also holds for RELU BRNNs and BRNNs using RELU or RELU-LIKE activation functions. The proof of Theorem 26 is presented in Appendix F.5.

**Theorem 26.** *For any $T \in \mathbb{N}$ and $\varphi \in \{\text{RELU}\} \cup \text{RELU-LIKE}$, $w_{\min} \leq \{d_x, d_y, 2\}$ for $\varphi$ BRNNs to be dense in $L^p([0,1]^{d_x \times T}, \mathbb{R}^{d_y \times T})$.*

### F.3 PROOF OF THEOREM 24

#### F.3.1 PROOF OUTLINE FOR RELU RNNS

In this section, we show that for any past-dependent $f^* \in L^p([0,1]^{d_x \times T}, \mathbb{R}^{d_y \times T})$ and $\varepsilon > 0$, there exists a RELU RNN $f : [0,1]^{d_x \times T} \to \mathbb{R}^{d_y \times T}$ of width $\max\{d_x, d_y, 2\}$ such that

$$\|f - f^*\|_{p,p} \leq \varepsilon.$$

Without loss of generality, we restrict the codomain to $[0,1]^{d_y \times T}$. Then, since continuous functions in $C([0,1]^{d_x \times T}, \mathbb{R}^{d_y \times T})$ are dense in $L^p([0,1]^{d_x \times T}, \mathbb{R}^{d_y \times T})$ (Rudin, 1987), it suffices to prove the following statement: for any $\varepsilon > 0$, $f' \in C([0,1]^{d_x \times T}, [0,1]^{d_y \times T})$, there exists a RELU RNN $f : [0,1]^{d_x \times T} \to [0,1]^{d_y \times T}$ of width $\max\{d_x, d_y, 2\}$ satisfying

$$\|f' - f\|_{p,p} \leq \varepsilon.$$

We explicitly construct such RELU RNN $f$ using the coding scheme. To describe this, we present the following lemmas where the proofs of Lemmas 27–29 are presented in Appendices F.6–F.8, respectively.

**Lemma 27.** *Given $T \in \mathbb{N}$ and $\alpha, \beta > 0$, there exist disjoint measurable sets $\mathcal{T}_1, \ldots, \mathcal{T}_k \subset [0,1]^{d_x}$ and a RELU RNN $g^\dagger : [0,1]^{d_x \times T} \to \mathbb{R}^T$ of width $\max\{d_x, 2\}$ such that*

- $\mathrm{diam}(\mathcal{T}_i) \leq \alpha$ *for all $i \in [k]$,*

- $\mu_{d_x}\left(\bigcup_{i=1}^k \mathcal{T}_i\right) \geq 1 - \beta$, *and*

- *if $x \in [0,1]^{d_x \times T}$ satisfies $x[t] \in \mathcal{T}_{i_t}$ for all $t \in [T]$, then $g^\dagger(x)[t] = c_{i_t}$ for all $t \in [T]$, for some distinct $c_1, \ldots, c_k \in \mathbb{R}$.*

Lemma 27 states that for any $\alpha, \beta > 0$, there exist $\mathcal{T}_1, \ldots, \mathcal{T}_k$ satisfying properties in Lemma 27 and a RELU RNN $g^\dagger$ of width $\max\{d_x, 2\}$ that assigns distinct codewords to each token $x[t] \in \mathcal{T}_{i_t}$ for $t \in [T]$. However, unlike the fully-connected network case, the $t$-th token of the RNN output must be a function of $x[1:t]$. To encode information of $x[1:t]$, we introduce the following lemma.

**Lemma 28.** *Given $T \in \mathbb{N}$ and distinct $c_1, \ldots, c_k \in \mathbb{R}$, there exist*

- *distinct $a_j \in \mathbb{R}$ for all $j \in [k]^t$, and*

- *a RELU RNN $g^\ddagger : \mathbb{R}^{1 \times T} \to \mathbb{R}^{1 \times T}$ of width 2 such that for any $(c_{i_1}, \ldots, c_{i_T})$ with $i_1, \ldots, i_T \in [k]$*

$$g^\ddagger(x)[t] = a_{j_t}$$

*for all $t \in [T]$ where $j_t = (i_1, \ldots, i_t)$.*

Lemma 28 states that for any set of codewords $\{c_1, \ldots, c_k\}$, there exists an RNN encoder implemented by a RELU RNN $g^\ddagger$ of width 2 that maps any vector of codewords $(c_{i_1}, \ldots, c_{i_t})$ of length $t \in [T]$ into a single scalar codeword $a_{(i_1, \ldots, i_t)}$ with the following property: different vectors are mapped to distinct scalar codewords.

By Lemmas 27 and 28, one can observe that for any $\alpha, \beta > 0$, there exist $\mathcal{T}_1, \ldots, \mathcal{T}_k \subset [0,1]^{d_x}$, a RELU RNN $g'$ of width $\max\{d_x, 2\}$, and $a_j \in \mathbb{R}$ for all $j \in \bigcup_{t=1}^T [k]^t$ satisfying the following properties:

- $\mathrm{diam}(\mathcal{T}_i) \leq \alpha$ for all $i \in [k]$,
- $\mu_{d_x}\left(\bigcup_{i=1}^k \mathcal{T}_i\right) \geq 1 - \beta$,
- $a_j \neq a_{j'}$ if $j \neq j'$, and
- if $x \in [0,1]^{d_x \times T}$ satisfies $x[t] \in \mathcal{T}_{i_t}$ for all $t \in [T]$, then $g'(x)[t] = a_{(i_1, \ldots, i_t)}$.

Namely, if $x \in \mathcal{T}_{i_1} \times \cdots \times T_{i_T}$, then $g'(x)[t] = a_{(i_1, \ldots, i_t)}$.

We next construct a RELU RNN that maps each $(a_{(i_1)}, \ldots, a_{(i_1, \ldots, i_T)})$ to some $y \in \mathbb{R}^{d_y \times T}$ such that $y[t]$ approximates $f'(\mathcal{T}_{i_1} \times \cdots \times \mathcal{T}_{i_T})[t]$ for all $t \in [T]$.

**Lemma 29.** *Given $T \in \mathbb{N}$, $p \geq 1$, $\gamma > 0$, distinct $a_1, \ldots, a_m \in \mathbb{R}$, and $v_1, \ldots, v_m \in \mathbb{R}^{d_y}$, there exists a RELU RNN $h : \mathbb{R}^{1 \times T} \to [0,1]^{d_y \times T}$ of width $\max\{d_y, 2\}$ such that for any $x = (a_{j_1}, \ldots, a_{j_T})$ with $j_1, \ldots, j_T \in [m]$,*

$$\|h(x)[t] - v_{j_t}\|_p \leq \gamma$$

*for all $t \in [T]$.*

By combining Lemmas 27–29, one can observe that for any $\alpha, \beta, \gamma > 0$, there exist $\mathcal{T}_1, \ldots, \mathcal{T}_k \subset [0,1]^{d_x}$, a RELU RNN $f$ of width $\max\{d_x, d_y, 2\}$, and $v_j \in \mathbb{R}^{d_y}$ for all $j \in \bigcup_{t=1}^{T} [k]^t$ satisfying the following properties:

- $\mathrm{diam}(\mathcal{T}_i) \leq \alpha$ for all $i \in [k]$,

- $\mu_{d_x}\left(\bigcup_{i=1}^{k} \mathcal{T}_i\right) \geq 1 - \beta$,

- if $x \in [0,1]^{d_x \times T}$ satisfies $x[t] \in \mathcal{T}_{i_t}$ for all $t \in [T]$, then

$$\|f(x)[t] - f'(x)[t]\|_p \leq \omega_{(p,p),F,f'}\left(\alpha\sqrt{T}\right) + (d_y T^2 \beta)^{1/p} + T^{1/p}\gamma$$

  for all $t \in [T]$,[1] and

- if $x \in [0,1]^{d_x \times T}$ satisfies $x[t] \notin \mathcal{T}_{i_t}$ for some $t \in [T]$, then $f(x) \in [0,1]^{d_y \times T}$.

We will show that such RNN $f$ of width $\max\{d_x, d_y, 2\}$ satisfies $\|f - f'\|_{p,p} \leq \varepsilon$ under proper choices of $\alpha, \beta, \gamma > 0$.

### F.3.2 OUR CHOICES OF $\alpha, \beta, \gamma$ FOR RELU RNNS

We choose sufficiently small $\alpha > 0$ so that $\omega_{(p,p),F,f'}(\alpha\sqrt{T}) \leq \varepsilon/2^{1+1/p}$, $\beta = \varepsilon^p/(2d_y T^2)$, and $\gamma = \varepsilon/(2^{1+1/p}T^{1/p})$. For convenience, we use $\mathcal{T} \triangleq \bigcup_{i=1}^{k} \mathcal{T}_i$. Under this setup, we bound the error using the following inequality:

$$\|f - f'\|_{p,p}^p = \int_{[0,1]^{d_x \times T}} \|f'(x) - f(x)\|_{p,p}^p d\mu_{d_x T}$$

$$\leq T \times \sup_{1 \leq t \leq T} \int_{[0,1]^{d_x \times T} \setminus \mathcal{T}^T} \|f'(x)[t] - f(x)[t]\|_p^p d\mu_{d_x T} + \int_{\mathcal{T}^T} \|f'(x) - f(x)\|_{p,p}^p d\mu_{d_x T}. \quad (13)$$

We first bound the first term in RHS of Eq. (13). Note that both $f$ and $f'$ have codomain is $[0,1]^{d_y}$.

$$T \times \sup_{1 \leq t \leq T} \int_{[0,1]^{d_x \times T} \setminus \mathcal{T}^T} \|f'(x) - f(x)\|_{p,p}^p d\mu_{d_x T}$$

$$= T \times d_y \mu_{d_x T}\left(\bigcup_{j=1}^{T} [0,1]^{d_x \times (T-j)} \times ([0,1]^{d_x} \setminus \mathcal{T}) \times [0,1]^{d_x \times (j-1)}\right)$$

$$\leq T d_y \sum_{j=1}^{T} (1 - \mu_{d_x}(\mathcal{T})) \leq d_y T^2 \beta \leq \varepsilon^p/2. \quad (14)$$

---

[1] $\omega_{(p,p),F,f'}$ denotes the modulus of continuity of $f'$ in the $L_{p,p}$-norm and Frobenius-norm: $\|f'(x) - f'(x')\|_{p,p} \leq \omega_{(p,p),F,f'}(\|x - x'\|_F)$ for all $x, x' \in [0,1]^{d_x \times T}$.

We next bound the second term in RHS of Eq. (13) using Minkowski's inequality as follows:

$$\int_{\mathcal{T}^T} \|f'(x) - f(x)\|_{p,p}^p d\mu_{dxT}$$

$$= \sum_{i_1,\ldots,i_T \in [k]} \int_{\prod_{s=1}^T \mathcal{T}_{i_s}} \|f(x) - f'(x)\|_{p,p}^p d\mu_{dxT}$$

$$\leq \sum_{i_1,\ldots,i_T \in [k]} \int_{\prod_{s=1}^T \mathcal{T}_{i_s}} (\|f'(z_{j_T}) - f'(x)\|_{p,p} + \|f(x) - f'(z_{j_T})\|_{p,p})^p d\mu_{dxT}$$

$$\leq \sum_{i_1,\ldots,i_T \in [k]} \left[ \left( \int_{\prod_{s=1}^T \mathcal{T}_{i_s}} \|f'(z_{j_T}) - f'(x)\|_{p,p}^p d\mu_{dxT} \right)^{1/p} \right.$$

$$\left. + \left( \int_{\prod_{s=1}^T \mathcal{T}_{i_s}} \|f(x) - f'(z_{j_T})\|_{p,p}^p d\mu_{dxT} \right)^{1/p} \right]^p$$

$$\leq \left[ \left( \sum_{i_1,\ldots,i_T \in [k]} \int_{\prod_{s=1}^T \mathcal{T}_{i_s}} \|f'(z_{j_T}) - f'(x)\|_{p,p}^p d\mu_{dxT} \right)^{1/p} \right.$$

$$\left. + \left( \sum_{i_1,\ldots,i_T \in [k]} \int_{\prod_{s=1}^T \mathcal{T}_{i_s}} \|f(x) - f'(z_{j_T})\|_{p,p}^p d\mu_{dxT} \right)^{1/p} \right]^p$$

$$\leq \left[ \left( \sum_{i_1,\ldots,i_T \in [k]} \int_{\prod_{s=1}^T \mathcal{T}_{i_s}} \left( \omega_{(p,p),F,f'} (\|z_{j_T} - x\|_F) \right)^p d\mu_{dxT} \right)^{1/p} \right.$$

$$\left. + \left( \sum_{i_1,\ldots,i_T \in [k]} \int_{\prod_{s=1}^T \mathcal{T}_{i_s}} \sum_{t \in [T]} \|f(x)[t] - v_{j_t}\|_p^p d\mu_{dxT} \right)^{1/p} \right]^p$$

$$\leq \left[ \left( \sum_{i_1,\ldots,i_T \in [k]} \int_{\prod_{s=1}^T \mathcal{T}_{i_s}} \left( \omega_{(p,p),F,f'} \left( \alpha\sqrt{T} \right) \right)^p d\mu_{dxT} \right)^{1/p} \right.$$

$$\left. + \left( \sum_{i_1,\ldots,i_T \in [k]} \int_{\prod_{s=1}^T \mathcal{T}_{i_s}} T\gamma^p d\mu_{dxT} \right)^{1/p} \right]^p$$

$$\leq \left( \omega_{(p,p),F,f'} \left( \alpha\sqrt{T} \right) + T^{1/p}\gamma \right)^p \leq \varepsilon^p/2 \tag{15}$$

where $z_{j_T} \in \prod_{s=1}^T \mathcal{T}_{i_s}$ for all $i_s \in [k]$. The second term in the above bound used our construction of $v_{j_t}$ and $f = h \circ g^{\ddagger} \circ g^{\dagger}$. By combining Eqs. (13)–(15), we have

$$\|f - f^*\|_{p,p} \leq \frac{\varepsilon}{2} + \frac{\varepsilon}{2} \leq \varepsilon.$$

This completes the proof of Theorem 24.

## F.4 PROOF OF THEOREM 25

In this section, we prove that any RELU RNN (or BRNN) $f$ can be approximated by an RNN (or BRNN) $g$ of the same width using any of RELU-LIKE activation functions, within any uniform error. Note that a RELU RNN $f$ of width $w$ does not imply $f$ is a RELU network with width $w$. Hence, we need the following extended definition for analysis of RELU RNN.

Given an activation function $\sigma : \mathbb{R} \to \mathbb{R}$, we define a $\sigma$ *token-network* as follows:

$$f(x_1, x_2, \ldots, x_T) \triangleq \psi_L \circ \psi_{L-1} \circ \cdots \circ \psi_2 \circ \psi_1, \tag{16}$$

where $\psi_\ell$ is one of the following operations.

- applying affine transformation $t(\cdot)$ on $k$-th token:

$$\psi_{t(\cdot),k}(x_1, x_2, \ldots, x_T) \triangleq (x_1, x_2, \ldots, t(x_k), \ldots, x_T)$$

  where $W_t \in \mathbb{R}^{d_t \times d}$, $b_t \in \mathbb{R}^{d_t}$, $x_k \in \mathbb{R}^d$ and $t(x) = W_t x + b_t$ for some $d, d_t \in \mathbb{N}$.
- element-wise $\sigma$ activation on $k$-th token:

$$\psi_{\sigma,k}(x_1, x_2, \ldots, x_T) \triangleq (x_1, x_2, \ldots, \phi_\sigma(x_k), \ldots, x_T)$$

  where $\phi_\sigma$ is an element-wise activation function.
- copying the $k$-th token $\psi_c$ to a new token:

$$\psi_{c,k}(x_1, x_2, \ldots, x_T) \triangleq (x_1, x_2, \ldots, x_T, x_k)$$

- adding two tokens with the same dimension into a new token:

$$\psi_{s,k,l}(x_1, x_2, \ldots, x_T) \triangleq (x_1, x_2, \ldots, x_T, x_k + x_l)$$

- deleting $k$-th token:

$$\psi_{d,k}(x_1, x_2, \ldots, x_T) \triangleq (x_1, x_2, \ldots x_{k-1}, x_{k+1}, \ldots, x_T)$$

The width $w$ of a $\sigma$ token-network $f$ is defined as the maximum of input/output dimensions of affine transformations $t$ that are applied in $f$. Remark that $\sigma$ RNN (or BRNN) of width $w$ is a $\sigma$ token-network with width $w$. If we define $\|(x_1, \ldots, x_T)\|_\infty \triangleq \sup_{t \in [T]} \|x_t\|_\infty$, we can apply the same method as in the proof of Lemma 8 in Appendix E. When $\psi$ is either copying or deleting, then

$$\|\psi(X) - \psi(Y)\|_\infty \leq \|X - Y\|_\infty$$

and when $\psi$ is adding tokens, then

$$\|\psi(X) - \psi(Y)\|_\infty \leq 2\|X - Y\|_\infty$$

above error bound holds.

So for a given RELU token-network $f$, the following inequality holds for every $\psi$ in $f$:

$$\|\psi(X) - \psi(Y)\|_\infty \leq \max\{2, M\}\|X - Y\|_\infty$$

where $M$ is maximum value of norm of affine transformation $\|W_t\|_\infty$ in $f$. Therefore, using the identical method as in Appendix E, we are able to construct RELU-LIKE token-network $g$ such that

$$\|f(X) - g(X)\|_\infty \leq \varepsilon.$$

Since uniform convergence of functions in compact domain implies $p$-norm convergence, we are able to extend the result of RELU RNNs to RNNs using RELU-LIKE activation functions, hence the proof of Theorem 25 is completed.

### F.5 PROOF OF THEOREM 26

In this proof, we follow the discussion in Appendix F.3. We use Lemmas 27 and 29 and a modified version of Lemma 28 to construct encoder and decoder BRNNs. The modified lemma is as follows:

**Lemma 30.** *Given $T \in \mathbb{N}$ and distinct $c_1, \ldots, c_k \in \mathbb{R}$, there exist*

- *distinct $a_{j_t, \bar{j}_t} \in \mathbb{R}$ for all $t \in [T]$, $j_t \in [k]^t$, and $\bar{j}_t \in [k]^{T-t+1}$, and*

- *a RELU RNN $g^\ddagger : \mathbb{R}^{1 \times T} \to \mathbb{R}^{1 \times T}$ of width 2 such that for any $x := (c_{i_1}, \ldots, c_{i_T})$ with $i_1, \ldots, i_T \in [k]$*

$$g^{\ddagger}(x)[t] = a_{j_t, \bar{j}_t}$$

*for all $t \in [T]$ where $j_t = (i_1, \ldots, i_t)$ and $\bar{j}_t = (i_t, \ldots, i_T)$.*

Note that $a_{j_t}$ in Lemma 28 only depends on the past whereas $a_{j_t, \bar{j}_t}$ in Lemma 30 depends both on the past and future. The proof of Lemma 30 is provided in Appendix F.9.

Now, we are ready to construct our BRNN model $f$. First, Lemma 27 ensures that there exist $\mathcal{T}_1, \ldots, \mathcal{T}_k \subset [0,1]^{d_x}$ and a RELU RNN $g^{\dagger}$ of width $\max\{d_x, 2\}$. Also from Lemma 30, there exist $a_{j, \bar{j}} \in \mathbb{R}$ for all $j, \bar{j} \in \bigcup_{t=1}^{T} [k]^t \times [k]^{T-t+1}, j[t] = \bar{j}[0]$, and a RELU RNN $g^{\ddagger}$ satisfying the following properties:

- $\mathsf{diam}(\mathcal{T}_i) \le \alpha$ for all $i \in [k]$,
- $\mu_{d_x}\left(\bigcup_{i=1}^{k} \mathcal{T}_i\right) \ge 1 - \beta$,
- $a_{j, \bar{j}} \ne a_{j', \bar{j}'}$ if $(j, \bar{j}) \ne (j', \bar{j}')$, and
- if $x \in [0,1]^{d_x \times T}$ satisfies $x[t] \in \mathcal{T}_{i_t}$ for all $t \in [T]$, then $g^{\ddagger} \circ g^{\dagger}(x)[t] = a_{(i_1, \ldots, i_t), (i_t, \ldots, i_T)}$.

Namely, if $x \in \mathcal{T}_{i_1} \times \cdots \times T_{i_T}$, then $g^{\ddagger} \circ g^{\dagger}(x)[t] = a_{(i_1, \ldots, i_t), (i_t, \ldots, i_T)}$. Now, for a given target function $f^* \in L^p([0,1]^{d_x \times T}, \mathbb{R}^{d_y \times T})$, we choose $z_{j_T} \in \prod_{s=1}^{T} \mathcal{T}_{i_s}$ as in Appendix F.3. Then, we define:

$$v_{j_t, \bar{j}_t} = f'(z_{j_T})[t].$$

By Lemma 29, one can construct decoder $h$ with respect to $v_{j_t, \bar{j}_t}$ such that

$$\|h(a_{j_t, \bar{j}_t}) - v_{j_t, \bar{j}_t}\|_p \le \gamma.$$

Note that $h$ is a token-wise function that can be constructed by a RELU BRNN. Then, the error bound in Appendix F.3 indicates that for any $\varepsilon > 0$, we have

$$\|f^* - f\|_{p,p} \le \epsilon$$

where $f = h \circ g^{\ddagger} \circ g^{\dagger}$.

Hence, using the extension of RELU token-network to RELU-LIKE token-network as in Appendix F.4 completes the statement of Theorem 26.

## F.6 PROOF OF LEMMA 27

To this end, we recall the statement of Lemma 5: For any $\alpha, \beta > 0$, there exist disjoint measurable sets $\mathcal{T}_1, \ldots, \mathcal{T}_k \subset [0,1]^{d_x}$ and a RELU network $g : \mathbb{R}^{d_x} \to \mathbb{R}$ of width $\max\{d_x, 2\}$ such that

- $\mathsf{diam}(\mathcal{T}_i) \le \alpha$ for all $i \in [k]$,
- $\mu_{d_x}\left(\bigcup_{i=1}^{k} \mathcal{T}_i\right) \ge 1 - \beta$, and
- $g(\mathcal{T}_i) = \{c_i\}$ for all $i \in [k]$, for some distinct $c_1, \ldots, c_k \in \mathbb{R}$.

Therefore, the statement of Lemma 27 directly follows from token-wise implementing a RELU network $g$, that is, $g^{\dagger}(x) = (g(x[1]), \ldots, g(x[T]))$ for any $x \in [0,1]^{d_x \times T}$.

## F.7 PROOF OF LEMMA 28

In this section, we explicitly construct a RELU RNN $f : \mathbb{R}^{1 \times T} \to \mathbb{R}^{1 \times T}$ satisfying the statement of Lemma 28. Without loss of generality, we assume that the distinct points $c_1, \ldots, c_k$ are contained in $[0,1]$. Then, we quantize each distinct point in the binary representation using a token-wise RELU network $g : \mathbb{R} \to \mathbb{R}$ of width 2 such that for any $K \in \mathbb{N}, \delta > 0$, and all $x \in [0,1] \setminus \mathcal{D}_{K, \delta}$,

$$g(x) = q_K(x)$$

where $\mathcal{D}_{K, \delta} = \bigcup_{i=1}^{2^K-1} (i \times 2^{-K} - \delta, i \times 2^{-K})$. The existence of such $g$ is ensured from Lemma 16.

Here, we recall the definition of the quantization function. A quantization function $q_K : [0, 1] \to \mathcal{C}_K$ for $K \in \mathbb{N}$ and $\mathcal{C}_K \triangleq \{0, 2^{-K}, 2 \times 2^{-K}, 3 \times 2^{-K}, \dots, 1 - 2^{-K}\}$ is defined as

$$q_K(x) = \max\{c \in \mathcal{C}_K : c \leq x\}.$$

One can observe that $g$ preserves the first $K$-bits in the binary representation and discards the rest bits. Nonetheless, we can ignore the information loss, which is the duplication of points, incurred from the quantization by choosing sufficiently large $K$ and small $\delta$ so that $2^{-(K+1)} < \inf_{i \neq j \in [k]} |c_i - c_j|$ and $\delta < 2^{-(K+2)}$.

Subsequently, we implement a RNN cell $\vec{R} : \mathbb{R}^{1 \times T} \to \mathbb{R}^{1 \times T}$ of width 1 defined as follows:

$$\vec{R}(x)[t+1] = \text{RELU}(2^{-K} \times \vec{R}(x)[t] + x[t+1]).$$

Then, such $\vec{R}$ successfully accumulates $(d_x \times t)$-bits of the binary representation of $x[1 : t] \in \mathbb{R}^{d_x \times t}$ for each $t \in [T]$ since $g(\{c_1, \dots, c_k\}) \subset [0, 1]$.

Lastly, let $G : \mathbb{R}^{1 \times T} \to \mathbb{R}^{1 \times T}$ be a RELU RNN of width 2 such that $G(x) = (g(x[1]), \dots, g(x[T]))$ for all $x \in \mathbb{R}^{d_x \times T}$. Then, the RELU RNN $f = \vec{R} \circ G$ of width 2 completes the proof of Lemma 28.

### F.8 PROOF OF LEMMA 29

From Lemma 6, there exits a RELU network $g : \mathbb{R} \to [0, 1]^{d_y}$ of width $\max\{d_y, 2\}$ such that for any $p \geq 1$, $\gamma > 0$, $m \in \mathbb{N}$, distinct $a_1, \dots, a_m \in \mathbb{R}$, and $v_1, \dots, v_m \in \mathbb{R}^{d_y}$,

$$\|g(a_i) - v_i\|_p \leq \gamma$$

for all $i \in [m]$. Therefore, the statement of Lemma 29 directly follows by token-wise implementing such RELU network $g$, that is, $h(x) = (g(a_{j_1}), \dots, g(a_{j_T}))$ for any $j_1, \dots, j_T \in [m]$.

### F.9 PROOF OF LEMMA 30

In this section, we follow the similar arguments as in Appendix F.7 to construct RELU BRNN $f : \mathbb{R}^{1 \times T} \to \mathbb{R}^{1 \times T}$ satisfying the statement of Lemma 30. Again, we assume that the distinct points $c_1, \dots, c_k$ are contained in $[0, 1]$. Then, there exists a token-wise RELU network $g : \mathbb{R} \to \mathbb{R}$ of width 2 such that for any $K \in \mathbb{N}$, $\delta > 0$, and all $x \in [0, 1] \setminus \mathcal{D}_{K,\delta}$,

$$g(x) = q_K(x)$$

where $\mathcal{D}_{K,\delta}$ and quantization function $q_K(x)$ is defined in Appendix F.7. Next, choose the same precision $K$ and small enough $\delta$ that satisfies $2^{-(K+1)} < \inf_{i \neq j \in [k]} |c_i - c_j|$ and $\delta < 2^{-(K+2)}$.

We now implement a BRNN cell $\vec{\bar{R}} : \mathbb{R}^{1 \times T} \to \mathbb{R}^{1 \times T}$ of width 1 defined as follows:

$$\vec{R}(x)[t+1] = \text{RELU}(2^{-K} \times \vec{R}(x)[t] + x[t+1]),$$
$$\bar{R}(x)[t-1] = \text{RELU}(2^{-K} \times \bar{R}(x)[t] + 2^{-KT}x[t-1]),$$
$$\vec{\bar{R}}(x)[t] = \vec{R}(x)[t] + \bar{R}(x)[t].$$

Then, $\vec{\bar{R}}$ successfully accumulates $(d_x \times t)$-bits for $x[1 : t]$ and $(d_x \times (T - t + 1))$-bits for $x[t : T]$. Note that $2^{-KT}x[t-1]$ in $\bar{R}$ enables us to prevent overlapping of information from $\vec{R}$ by storing data bits in different positions.

Lastly, let $G : \mathbb{R}^{1 \times T} \to \mathbb{R}^{1 \times T}$ be a RELU BRNN of width 2 such that $G(x) = (g(x[1]), \dots, g(x[T]))$ for all $x \in \mathbb{R}^{d_x \times T}$. Then, the RELU BRNN $f = \vec{\bar{R}} \circ G$ of width 2 completes the proof of Lemma 30.

