# OpenReview forum: "Minimum width for universal approximation using ReLU networks on compact domain"
_ICLR.cc/2024/Conference — ICLR 2024 poster_

### Official Review · Reviewer_FUZf · 2023-10-12

**Soundness:** 3 good
**Presentation:** 3 good
**Contribution:** 2 fair
**Rating:** 8
**Confidence:** 5

**Summary:**

This work studies the minimum width required for universal approximation ability of neural networks, under general settings with varying norm, input/output dimensions and activation functions. In particular, this work generalizes the result of Cai 2023, showing that the minimum width is exactly $\max(d_x,d_y,2)$ for neural networks with ReLU-class activations to approximate $L^p$ functions from a compact set of $\mathbb{R}^{d_x}$ to $\mathbb{R}^{d_y}$. Then it's shown that when uniform approximation is considered, the minimum width is at least $d_y+1$ when $d_x\le d_y\le 2d_x$, implying a dichotomy between $L^p$ and uniform approximation.

**Strengths:**

**Solid results:** the theoretical results make a solid contribution to the understanding of the universal approximation ability of neural networks.

**Wide coverage:** Theorem 2 holds for a wide class of ReLU-like activations, while previous works mostly consider only the most representative ReLU. Though such result is well expected due to the similarity between these activations, and indeed the proof is based on a simple reduction, it is non-trivial and such generality is valuable.

**Tight bounds:** since the initial work of Lu et al 2017 which achieves the $d_x+4$ bound in the particular setting with $L_1$ norm, $d_y=1$ and ReLU activation, there has been a line of works on sharpening the bound itself, and generalizing the setting to other norms and activations. This work finally presents an exact characterization of the minimum width for general $L^p$ norm and a wide class of ReLU-like activations.

**Weaknesses:**

**Questionable significance:** though this work makes a solid contribution to a tight characterization of the minimum width for universal approximation which is certainly valuable for our theoretical understanding, in my opinion, the mission itself to improve upon previous results is not so significant. The gap between known upper and lower bounds is merely an additive constant, and a similar tight result was achieved in Cai 2023 for the special case of Leaky-ReLU.

**Questions:**

As a side note, the separation between the whole Euclidean space and compact subset (equivalently, $L^p$ versus uniform) was noticed even before Wang and Qu 2022. In a technical report of Lu [1], it's shown that any two-layer ReLU network must incur $\int_{\mathbb{R}^d}|f|$ error to approximate an integrable function $f$, a sharp contrast to the case when the integral is deployed on a compact set. Their argument is very simple and may be potentially helpful for explaining the intuition of such separation results.

[1], A note on the representation power of ghhs, Zhou Lu, arXiv preprint 2101.11286

---

> ### Author Response · Authors · 2023-11-19
> **Response to Reviewer FUZf**
>
> We thank the reviewer for their positive evaluation and valuable feedback. We address all comments of the reviewer, and provide pointers to the corresponding updates in the revised draft. All the updates are color-coded in the revised version.
>
>
> **A similar tight result was achieved in Cai 2023 for the special case of Leaky-ReLU.**
>
> We would like to emphasize that the construction for achieving the tight minimum width of Leaky-ReLU networks for $L^p$ approximation by Cai (2023) does not generalize to ReLU networks. Cai (2023) constructed a Leaky-ReLU networks of width $\max\\{d_x,d_y,2\\}$ by combining two observations: (1) $L^p$ functions can be approximated by neural ODEs (Li et al., 2022) and (2) neural ODEs can be approximated by Leaky-RELU networks of width $\max\\{d_x,d_y,2\\}$ (Duan et al., 2022). However, the second observation requires the strict monotonicity of Leaky-ReLU networks, which does not extend to ReLU ones. To bypass this issue and show the tight minimum width for ReLU networks, we propose a completely different approach based on the characteristics of ReLU networks (Lemmas 5 and 7), which further extends to other (possibly non-monotone) ReLU-Like activation functions. We have clarified this difference in the revised draft (page 4 in Section 3).
>
> **The gap between known upper and lower bounds is merely an additive constant.**
>
> Although our results improve constant factors of existing bounds on the minimum width, we believe that our results are important contributions to the expressive power of neural networks. As the reviewer pointed out, our Theorems 1 and 2 provide a tight minimum width of networks using various activation functions for $L^p$ approximation. We think such tight characterization enables us to understand how the expressive power of neural networks can be affected by problem setups: for example, our Theorems 1 and 2 show dichotomy between the minimum widths for $L^p$ approximation on the whole Euclidean space and compact sets. To our knowledge, such a separation result was previously unknown in terms of the minimum width.
>
> Theorem 3 also generalizes dichotomy between $L^p$ and uniform approximation to general activation functions and general input/output dimensions. Furthermore, we recently found that *Theorem 3 is tight when $2d_x=d_y$* for Leaky-ReLU networks, together with the matching upper bound $\max\\{2d_x+1,d_y\\}$ [Hwang, 2023]. We think our tight minimum width results can help better understand the universal approximability of deep and narrow neural networks.
>
> We have added the new tight minimum width result using Theorem 3 to the revised draft (page 3 in Section 1.2 and page 5 in Section 3).
>
> [Hwang, 2023] Minimum Width for Deep, Narrow MLP: A Diffeomorphism Approach, arXiv preprint 2308.15873
>
> **On separation between the whole Euclidean space and compact subset.**
>
> We thank the reviewer for providing a reference [Lu, 2021] that we were not aware of. We have cited this work in the discussion about the separation between the whole Euclidean space and compact subset in our revised draft (page 4 in Section 3).
>
> [Lu, 2021] A note on the representation power of ghhs, Zhou Lu, arXiv preprint 2101.11286
>
>
> We would be happy to clarify any concerns or answer any questions that may come up during the discussion period.

---

> > ### Comment · Reviewer_FUZf · 2023-11-22
> >
> > Thanks for your reply. Now I see the technical novelty over Cai (2023), and since ReLU is a more fundamental case than Leaky-ReLU, I have increased the score accordingly.

---

### Official Review · Reviewer_fZsP · 2023-10-28

**Soundness:** 4 excellent
**Presentation:** 4 excellent
**Contribution:** 3 good
**Rating:** 8
**Confidence:** 5

**Summary:**

The authors, in a sense, improve on the available results quantifying the minimal widths required for a class of deep but narrow MLPs to be universal in $C([0,1]^{d_X},\mathbb{R}^{d_Y})$.    The result is very interesting, and of a technical nature; namely, they show that minimal width can (but surprisingly and not shockingly) be improved when only considering approximation of functions on the cube and not the entire domain $\mathbb{R}^{d_X}$.

**Strengths:**

The results are interesting, especially for those studying the fundamental limits of MLPs and their approximation theory. The presentation is clear and the proofs are mathematically correct.

Though one may foreseeable argue that the contribution is of a technical nature, these results answer fundamental questions at the core of the validity of practical MLP implementations.

In short, I think these results should definitely be published :)

**Weaknesses:**

The paper is rigorous and I do not see any weaknesses; with one exception:

- Can the authors please add more details and rigor in the construction in Lemma 6's proof.  I know a lot of it is drawn from another paper's lemmata but it would be better to have it explicit and self contained.  Right now it is not even a proof but a proof sketch/recipe.

**Questions:**

** 1) Impact of metric entropy on minimal width?**

Fix a compact subset $X$ of $\mathbb{R}^{d_X}$, non-empty.  Suppose that we are looking for universal approximators in $C(X,\mathbb{R}^{d_Y})$  implementable by MLPs with ReLU-Like activation function and of bounded widths.  How do you expect that the metric entropy/capacity of $X$ will impact the minimum width?


For instance, if $X=\{x_0\}$ is a singleton and $d_Y=1$, then clearly the width $\min\{d_X,d_Y,2\}$ is suboptimal since the class since the class
$$
\{ x\mapsto a\operatorname{ReLU}(1\cdot (x+b)):\, a\in \mathbb{R} ,\, b:= -x_0 + 1\}
$$
is universal in $C(X,\mathbb{R})$.  So I guess there is room for improvement for general $X$.  (The same question applies to the case where $d_X=0$ and $d_Y=1$, in which case the minimum width is
$$
1 < \max\{d_X,d_Y,2\}=\max\{0,1,2\}=2.
$$


----

What's your intuition on how the metric entropy of $X$ appears into the estimate?


I thought about the case where $X=\{-1,1\}$ but minimal width seems to apply there also.  What am I missing?



----

** 2) Why not note more general implications?**

Perhaps I missed it, but it could be worth noting that your results also imply the minimal widths for universality/density in $C(\mathbb{R}^{d_X},\mathbb{R}^{d_Y})$ in the topology of uniform convergence on compact sets.  This is because of the extension and normalization arguments as in the proof of Proposition 3.10 [1] or in the proof of Proposition 53 [2], which allows one reduce the problem of universality in $C([0,1]^{d_X},\mathbb{R}^{d_Y})$.  I.e. using either of the Tiezte or McShane extension theorems


** 3) Improving Minimal Width Estimates for general nonlinearities**

In [5], the authors just recently showed that most MLPs with standard and continuous activation functions can approximately implement and MLP with ReLU activation function using roughly the same depth, width, and number of parameters.  I was wondering, unless I am missing something, why not use their results to sharpen your statement for general continuous activation functions?

- References -

[1] Acciaio, Beatrice, Anastasis Kratsios, and Gudmund Pammer. "Designing universal causal deep learning models: The geometric (Hyper) transformer." Mathematical Finance (2023).

[2] Kratsios, Anastasis, and Léonie Papon. "Universal approximation theorems for differentiable geometric deep learning." The Journal of Machine Learning Research 23, no. 1 (2022): 8896-8968.

[3] Arenas, Francisco García, and María Luz Puertas. "Tietze's extension theorem." Divulgaciones Matemáticas 10, no. 1 (2002): 63-78.

[4] Beer, Gerald. "McShane’s extension theorem revisited." Vietnam Journal of Mathematics 48, no. 2 (2020): 237-246.

[5] Zhang, Shijun, Jianfeng Lu, and Hongkai Zhao. "Deep Network Approximation: Beyond ReLU to Diverse Activation Functions." arXiv preprint arXiv:2307.06555 (2023).

---

> ### Author Response · Authors · 2023-11-19
> **Response to Reviewer fZsP**
>
> We thank the reviewer for their positive evaluation and thoughtful feedback. We address all comments of the reviewer, and provide pointers to the corresponding updates in the revised draft. All the updates are color-coded in the revised version.
>
> **More details and rigor in the construction in Lemma 6's proof.**
>
> We thank the reviewer for pointing this out. Following the reviewer’s comment, we have added details to the proof of Lemma 6 in the revised draft (pages 14-16 in Appendix B.4).
>
> **Impact of metric entropy on minimal width.**
>
> We are not sure if we correctly understand the definition of the metric entropy that you were considering, but we try to answer with the following definition, which appears in the covering argument in statistics: $\log N(X,\delta)$ where $N(X,\delta)$ denotes the minimum number of $\delta$-balls that can cover $X$ (i.e., the logarithm of the covering number). We consider $L^p$ approximation in this answer. If this problem setup is not what the reviewer was considering, please let us know.
>
> As the reviewer pointed out, any map from a single point (of the zero metric entropy) to a real number can be exactly represented by a ReLU network of width one. We think this observation extends to any domain of zero Lebesgue measure (including any finite sets) since we can ignore such sets in $L^p$ approximation; ignoring them incurs zero $L^p$ error. Under this observation, one interesting question we think is that “what if our error use a different measure, other than the Lebesgue one?” For example, the Cantor set $\mathcal C$ has zero Lebesgue measure but can have non-zero $d$-dimensional Hausdorff measure for $d<\log(2)/\log(3)$. In such a case, for universal approximation, we expect that ReLU networks of width one are insufficient since they can only represent monotone functions. Here, since the value $\log(2)/\log(3)$ is the Minkowski dimension of the Cantor set defined as $d_M(\mathcal C):=\lim_{\delta\to0^+} (\log N(\mathcal C,\delta))/(\log(1/\delta))$, one may find a connection between the metric entropy and the minimum width for universal approximation under a Hausdorff measure. We believe investigating the universal approximation property of neural networks under various measures is an interesting future research direction.
>
> **More general implications.**
>
> We thank the reviewer for this suggestion. As the reviewer pointed out, all of our results hold for an arbitrary compact domain in the Euclidean space. Using the extension lemmas provided by the reviewer, we have made this point clearer in the revised draft (page 5 in Section 3).
>
> **Improving minimal width estimates for general nonlinearities.**
>
> We appreciate the reviewer for this suggestion. [Zhang et al., 2023] approximate ReLU networks using a network using a class of activation functions by scaling the width and depth of a network with multiplicative factors 3 and 2. Applying this result to our Theorem 1 gives us an upper bound $3\max\\{d_x,d_y,2\\}$ on the minimum width. However, we found that this bound exceeds the known upper bound $\max\\{d_x+2,d_y+1\\}$ in Theorem 4 by Park et al. (2021).
>
> On the other hand, our results easily extend to other network architectures: recurrent neural networks (RNNs) and bidirectional recurrent neural networks (BRNNs). For these networks, we consider universally approximating the space of $L^p$ functions that maps a length $T$ sequence of $d_x$-dimensional vectors to a length $T$ sequence of $d_y$-dimensional vectors, denoted by $L^p([0,1]^{d_x\times T},\mathbb R^{d_y\times T})$. Since the $t$-th output of RNN is always a function of the first to $t$-th inputs, for RNNs, we consider universally approximating the space of such functions, called *past-dependent* $L^p([0,1]^{d_x\times T},\mathbb R^{d_y\times T})$, while we consider $L^p([0,1]^{d_x\times T},\mathbb R^{d_y\times T})$ for BRNNs.
>
> We recently showed that the minimum width of RNNs to be dense in past-dependent $L^p([0,1]^{d_x\times T},\mathbb R^{d_y\times T})$ is exactly $\max\\{d_x,d_y,2\\}$ if the activation function is one of ReLU, SOFTPLUS, Leaky-RELU, ELU, CELU, SELU, GELU, SILU, and MISH; here GELU, SILU, and MISH require $d_x+d_y\ge3$. In addition, we also show that the minimum width of BRNNs to be dense in $L^p([0,1]^{d_x\times T},\mathbb R^{d_y\times T})$ is upper bounded by $\max\\{d_x,d_y,2\\}$ if the activation function is ReLU or in ReLU-Like. We have added these results with formal problem setups and proofs in the revised draft (pages 27-34 in Appendix F).
>
> [Zhang et al., 2023] Deep Network Approximation: Beyond ReLU to Diverse Activation Functions, arXiv preprint arXiv:2307.06555, 2023
>
>
> We would be happy to clarify any concerns or answer any questions that may come up during the discussion period.

---

> > ### Comment · Reviewer_fZsP · 2023-11-22
> > **Thanks**
> >
> > Dear Authors thank you for the clarifications, adding details to Lemma 6's proof, and for the very interesting discussion.  I think this is a very nice paper.

---

### Official Review · Reviewer_4yPi · 2023-10-29

**Soundness:** 3 good
**Presentation:** 3 good
**Contribution:** 2 fair
**Rating:** 5
**Confidence:** 4

**Summary:**

In this paper, the authors study the universal approximation problem of deep neural networks with unlimited depth. The main contribution of this paper is to derive that when the input domain and output domain are $[0,1]^{d_x}$ and $\mathbb R^{d_y}$ respectively, the minimum width of the universal approximation of neural networks for $L^p$ functions is equal to $\max(d_x,d_y,2)$, when the activation function is similar to RELU (e.g., RELU, GELU, SOFTPLUS). The authors also show that if the activation function is a continuous function that can be uniformly approximated by a sequence of continuous one-to-one functions, then the minimum width of the universal approximation of neural networks for continuous functions is at least $d_y+1$ if $d_x<d_y \leq 2d_x$.

**Strengths:**

Originality: The related works are adequately cited. The main results in this paper will certainly help us have a better understanding of the universal approximation property of deep neural networks from a theoretical way. I have checked the technique parts and found that the proofs are solid. One of the main results, which shows that there is a dichotomy between $L^p$ and uniform approximations for general activation functions and input/output dimensions, is a non-trivial extension of previous results in this field.

Quality: This paper is technically sound.

Clarity: This paper is clearly written and well organized. I find it easy to follow.

Significance: I think the results in this paper are not very significant, as explained below.

**Weaknesses:**

However, I have several concerns about the contribution of this paper. Firstly, the paper (Cai, 2023) already proved that the minimum width of the universal approximation of neural networks for $L^p$ functions is equal to $\max(d_x,d_y,2)$, when the activation function is Leaky-RELU. This paper only generalizes Leaky-RELU to RELU-LIKE activations (e.g., RELU, GELU, SOFTPLUS), and derives the same result. I think this makes the contribution of this paper incremental. Also, It would be more interesting if the authors could study the exact minimum width for more architectures used in practice. Furthermore, the technical part is not very deep and mostly based on the technical results from previous papers such as (Cai, 2023). In summary, I think this paper is a decent paper with some good results, but may not be suitable for the top conferences such as ICLR.

**Questions:**

As explained above, It would be more interesting if the authors could study the exact minimum width for more architectures used in practice.

---

> ### Author Response · Authors · 2023-11-19
> **Response to Reviewer 4yPi**
>
> We appreciate the reviewer for their time and effort to provide valuable comments. We address all comments of the reviewer, and provide pointers to the corresponding updates in the revised draft. All the updates are color-coded in the revised version.
>
> **This paper only generalizes Leaky-RELU to RELU-LIKE activations, and derives the same result. I think this makes the contribution of this paper incremental.**
>
> We believe that our contribution extending Leaky-ReLU to ReLU-Like activation is non-trivial: the existing proof for Leaky-ReLU networks does not extend to ReLU and ReLU-like activation functions (our Theorems 1 and 2). The proof of Cai (2023) for showing the upper bound on the minimum width $\max\\{d_x,d_y,2\\}$ consists of two parts: they (1) approximate $L^p$ functions via neural ODEs following (Li et al., 2022) and (2) approximate neural ODEs via Leaky-RELU networks of width $\max\\{d_x,d_y,2\\}$ by using results in (Duan et al., 2022). Here, the second part heavily relies on the strict monotonicity of Leaky-ReLU, and hence, does not generalize to ReLU. To bypass this issue and show the tight minimum width for ReLU networks, we propose a completely different proof technique (not using neural ODEs) utilizing properties of ReLU networks (Lemmas 5 and 7), which generalizes to (possibly non-monotone) ReLU-Like activation functions. We also note that our Theorems 1 and 2 first show the separation between the whole Euclidean space and compact subset for $L^p$ approximation, which was unknown up to our knowledge.
>
> Furthermore, we would like to emphasize that our Theorem 3 is also an important contribution. The best known lower bound on the minimum width for uniform approximation under general $d_x,d_y$ and general activation functions was $\max\\{d_x+1,d_y\\}$ (Johnson, 2019; Park et al., 2021). However, a few exceptional cases indicate that this bound can be improved: $d_y+1 > \max\\{d_x+1,d_y\\}$ is a tight lower bound for ReLU/Leaky-ReLU networks when $d_x=1$ and $d_y=2$ (Park et al., 2021; Cai, 2023). Our Theorem 3 improves existing lower bounds by showing that the minimum width for uniform approximation is at least $d_y+1$ if $d_x < d_y \le 2d_x$ for continuous activation functions that can be uniformly approximated by a sequence of continuous injections. As the reviewer pointed out, Theorem 3 also extends the dichotomy between and uniform approximations to general activation functions and input/output dimensions. In addition, we recently found that *Theorem 3 is tight when $d_y=2d_x$* for Leaky-ReLU networks, together with the matching upper bound $\max\\{2d_x+1,d_y\\}$ in [Hwang, 2023]. We think our tight minimum width results can help better understand universal approximability of deep and narrow neural networks.
>
> In the revised draft, we have clarified the difference between the existing Leaky-ReLU network result (Cai, 2023) and ours (page 4 in Section 3), and added the new tight minimum width result for uniform approximation using Theorem 3 (page 3 in Section 1.2 and page 5 in Section 3).
>
> [Hwang, 2023] Minimum Width for Deep, Narrow MLP: A Diffeomorphism Approach, arXiv preprint 2308.15873, 2023

---

> ### Author Response · Authors · 2023-11-19
> **Response to Reviewer 4yPi**
>
> **The technical part is not very deep and mostly based on the technical results from previous papers such as (Cai, 2023).**
>
> We believe that our technical results are not mostly based on previous papers such as (Cai, 2023). Theorem 1 (and Theorem 2) uses the lower bound $\max\\{d_x,d_y\\}$ from (Cai, 2023); however, this bound can be shown using rather straightforward arguments. If a network has width $d_x-1$, then it must have the form $g(Mx)$ for some continuous function $g:\mathbb R^{d_x-1}\to\mathbb R^{d_y}$ and $M \in \mathbb{R}^{(d_x-1) \times d_x}$. Hence, the network cannot use the full information of inputs and cannot universally approximate, e.g., consider approximating $\\|x\\|_2^2$. Likewise, if a network has width $d_y-1$, then it must have the form $Nh(x)$ for some continuous function $h:\mathbb R^{d_x}\to\mathbb R^{d_y-1}$ and $N\in\mathbb{R}^{d_y\times (d_y-1)}$ and cannot universally approximate, e.g., consider approximating a path that visits all vertices of a $d_y$-dimensional standard simplex along its edges. Combining these two arguments gives us the lower bound $\max\\{d_x,d_y\\}$.
>
> On the other hand, we believe our proof of the upper bound $\max\\{d_x,d_y,2\\}$ in Theorem 1, which we think the most critical part for showing the tight minimum width, is non-trivial and has technical novelty, especially compared to the lower bound. As we previously answered, prior result in (Cai, 2023) does not extend to ReLU (and ReLU-Like activation functions). We use a coding scheme; however, the existing coding-based ReLU network construction achieving the tight minimum width (Park et al., 2021) considers the whole Euclidean space and requires width at least $d_x+1$. Namely, existing results cannot directly show the tight minimum width in our problem setups: ReLU or ReLU-Like activation functions and a compact domain. We would like to also note that Theorem 3, which is also an important contribution of our submission, does not rely on any of previous technical results up to our knowledge.
>
> We have added this discussion to the revised draft (page 5 in Section 3).
>
> **Exact minimum width for more architectures used in practice.**
>
> We thank the reviewer for this interesting comment. Following the reviewer’s comment, we have explored other network architectures and found that our proof techniques can be used for bounding the minimum width of  recurrent neural networks (RNNs) and bidirectional RNNs (BRNNs) for $L^p$ approximation. Specifically, we consider universally approximating the space of $L^p$ functions that maps a length $T$ sequence of $d_x$-dimensional vectors to a length $T$ sequence of $d_y$-dimensional vectors, denoted by $L^p([0,1]^{d_x\times T},\mathbb R^{d_y\times T})$. Since the $t$-th output of RNN is always a function of the first to $t$-th inputs, for RNNs, we consider universally approximating the space of such functions, called *past-dependent* $L^p([0,1]^{d_x\times T},\mathbb R^{d_y\times T})$, while we consider $L^p([0,1]^{d_x\times T},\mathbb R^{d_y\times T})$ for BRNNs.
>
> We proved that the minimum width of RNNs to be dense in past-dependent $L^p([0,1]^{d_x\times T},\mathbb R^{d_y\times T})$ is exactly $\max\\{d_x,d_y,2\\}$ if the activation function is one of ReLU, SOFTPLUS, Leaky-RELU, ELU, CELU, SELU, GELU, SILU, and MISH; here GELU, SILU, and MISH require $d_x+d_y\ge3$ as in Theorem 2. Furthermore, we also show that the minimum width of BRNNs to be dense in $L^p([0,1]^{d_x\times T},\mathbb R^{d_y\times T})$ is upper bounded by $\max\\{d_x,d_y,2\\}$ if the activation function is ReLU or in ReLU-Like.
>
> We have added these new results with corresponding proofs and formal problem setups in the revised draft (pages 27-34 in Appendix F).
>
> We would be happy to clarify any concerns or answer any questions that may come up during the discussion period.

---

### Author Response · Authors · 2023-11-19
**Summary of revision**

Dear Reviewers and Area Chair,

We deeply appreciate your time and efforts to provide detailed comments on our submission. To best respond to your comments, we have revised our paper with additional results and clarifying contents. We listed our updates as below. All updates are color-coded in our revised draft.

- We have clarified the difference between our results (Theorems 1 and 2) and the existing result by Cai (2023) in Section 3 (Reviewers 4yPi and FUZf).
- We have included additional theoretical results on the minimum width of recurrent neural networks and bidirectional recurrent neural networks for $L^p$ approximation in Appendix F (Reviewers 4yPi and fZsP).
- We have added discussions about the tightness of Theorem 3 for Leaky-ReLU networks when $d_y=2d_x$ in Sections 1.2 and 3 (Reviewers 4yPi and FUZf).
- We have made the proof of Lemma 6 more rigorous and self-contained in Appendix B.4 (Reviewer fZsP).

Sincerely,\
Authors

---

### Meta-Review · Area_Chair_3TMa · 2023-12-22

**Metareview:**

The paper studies the minimum width required for universal approximation ability of neural networks, obtaining a new tight upper bound. The results are quite general, apply to several architectures, and  hold for a wide class of ReLU-like activations. The theoretical results presented in the paper seem like a strong contribution to the foundations of neural networks, improving our understanding of the expressivity of neural networks. In response to reviewer feedback, the authors added some additional results (e.g., on RNNs), updated the citations on the separation results between the whole Euclidean  space and compact subset, and clarified their contributions relative to Cai 2023.

**Justification For Why Not Higher Score:**

Some reviewers felt the results were incremental. I don’t think this is a valid criticism. Instead, universal approximation results for neural networks are often cited as evidence of their strength, but there are few, interesting applications of these results. In short, their impact is not that substantial.

**Justification For Why Not Lower Score:**

The results extend Cai 2023 results, and the obtained bounds apply to other ReLU-like activations (which previous work did not), and use novel mathematical approaches. In particular, the authors noted that Cai 2023 construction for achieving the tight min width for Leaky-ReLU networks does not generalize to ReLU networks.

---

### Decision · Program_Chairs · 2024-01-16

Accept (poster)